# Measuring mechanical anisotropy of the cornea with Brillouin microscopy

Amira M. Eltony [1], Peng Shao[1,3] & Seok-Hyun Yun [1,2]✉

Load-bearing tissues are typically fortified by networks of protein fibers, often with preferential orientations. This fiber structure imparts the tissues with direction-dependent mechanical properties optimized to support specific external loads. To accurately model and predict tissues' mechanical response, it is essential to characterize the anisotropy on a microstructural scale. Previously, it has been difficult to measure the mechanical properties of intact tissues noninvasively. Here, we use Brillouin optical microscopy to visualize and quantify the anisotropic mechanical properties of corneal tissues at different length scales. We derive the stiffness tensor for a lamellar network of collagen fibrils and use angle-resolved Brillouin measurements to determine the longitudinal stiffness coefficients (longitudinal moduli) describing the ex vivo porcine cornea as a transverse isotropic material. Lastly, we observe significant mechanical anisotropy of the human cornea in vivo, highlighting the potential for clinical applications of off-axis Brillouin microscopy.

[1] Harvard Medical School and Wellman Center for Photomedicine, Massachusetts General Hospital, Boston, MA 02114, USA. [2] Harvard-MIT Health Sciences and Technology, Cambridge, MA 02139, USA. [3] Present address: Reveal Surgical Inc., Montréal, QC H2N 1A4, Canada. ✉email: syun@hms.harvard.edu

B iological soft tissues are typically fiber-reinforced composites, consisting of a network of collagen and/or elastin fibers embedded in a hydrated matrix. In load-bearing regions, such as muscles, blood vessel walls, or the lining of the gastrointestinal tract, this microstructure imparts tissues with their unique combination of high flexibility and strength. Preferential fiber alignment is a natural consequence of the fiber structure conforming to resist the directional stresses and strains acting on a particular organ, resulting in anisotropic mechanical properties. Biaxial tensile testing has identified mechanical anisotropy in tissue samples excised from the lungs[1,2], the pulmonary arteries[3], the pericardium[4], the aorta[5], the retina[6], the vagina[7], and the skin[8,9] among others. The anisotropic behavior of these tissues is fine-tuned for their specific physiological functions. As such, fibrillar remodeling, resulting in changing tissue anisotropy, is often an indication of disease. There has been sustained interest in characterizing the mechanical anisotropy of soft tissues, particularly at the microstructural scale, in order to build accurate constitutive models which can be used to predict the tissues' response to mechanical loading[10].

Brillouin microscopy is an emerging optical technique for biomechanical characterization of cells and tissues[11]. The technique is based on inelastic light scattering from naturally occurring (spontaneous) or stimulated acoustic waves in the material being probed. The resulting frequency shift of the scattered light, or the Brillouin frequency shift, is related to the compressional acoustic wave speed and hence the longitudinal elastic properties of the material. By employing a focused laser beam, Brillouin microscopy is able to map elastic properties with optical-scale spatial resolution (~μm). Brillouin microscopy has been used to measure localized mechanical changes in the eyes of patients with the corneal disease keratoconus[12], and depth-dependent mechanical changes induced by collagen crosslinking (CXL)[13]. Brillouin scattering has revealed anisotropic properties of solid-state materials[14], silks[15], and plant stems[16], but to our knowledge, has not yet been used to analyze soft biological tissue such as the cornea.

The cornea has anisotropic mechanical properties because of preferential fibril alignment tangential to the surface of the eye. The mechanical strength, shape, and transparency of the cornea stem from its microstructure: an intricate lattice of collagen fibrils embedded in a gel matrix. Characterizing corneal anisotropy is important in predicting the response of the cornea to different mechanical stimuli. Measuring the degree of anisotropy may also be a useful indicator of changes in the cornea's collagen organization due to disease or following surgical intervention[17–19]. Some evidence of corneal anisotropy in the porcine eye has been obtained using ultrasound supersonic shear wave imaging[20], although with limited sensitivity and spatial resolution. Uniaxial tensile testing[21] and optical coherence elastography[22] have also been applied to the cornea, but were limited to comparing elasticity in tangential directions.

In this work, we directly visualize the alternating elastic properties of crisscrossing fiber layers (lamellae) in the porcine cornea using a high-resolution Brillouin microscope and confirm the interpretation of this pattern using transmission electron microscopy (TEM). We then measure the corresponding mechanical anisotropy in longitudinal modulus of the bulk porcine cornea using Brillouin microscopy which we find to be consistent with the microstructural-scale imaging. We derive a composite model for the lamellar network of collagen fibrils in the cornea and use our angle-resolved Brillouin measurements to determine the stiffness coefficients (longitudinal moduli) describing the transverse isotropic tissue. Finally, we analyze the angle-dependent Brillouin corneal maps of healthy human subjects in vivo and determine the mechanical anisotropy of the normal human cornea.

## Results

**Composite model of corneal anisotropy.** The stroma, which makes up the bulk of the cornea and contributes most to its biomechanical properties, is organized into layers ~2–3 μm-thick called lamellae. Within each lamella, collagen fibrils of ~25 nm-diameter are co-aligned tangentially to the corneal surface[23]. X-ray diffraction measurements suggest that collagen fibrils do not form a perfect crystalline lattice, but instead exhibit short-range order[24]. However, for the purpose of modeling, the fibril organization within an individual lamella can be well approximated by a pseudo-hexagonal lattice[25]. X-ray diffraction studies have established that the fibril diameter is 31 nm (humans) and 37 nm (pigs) and the interfibrillar Bragg spacing is 55 nm (humans) and 59 nm (pigs), corresponding to a fibril volume fraction $V^{(f)}$ of 0.22 (humans) and 0.28 (pigs)[25].

A single corneal lamella can be modeled as an aligned fiber composite, with the 1-direction oriented parallel to the fiber axis, and the 2- and 3-directions orthogonal to it (Fig. 1a). Since a collagen fibril can be assumed to be radially symmetric, it has transverse isotropic symmetry, and its stiffness tensor, $C^{(f)}$, has five independent coefficients (see Supplementary Materials). Likewise, the gel matrix surrounding the fibrils can also be described with a stiffness tensor, $C^{(m)}$. Assuming the fibers do not slip within the matrix under 1-directional loading (isostrain) and the stress is distributed equally across the fibers and matrix during 2- and 3-directional loading (isostress), the stiffness coefficients along the 1, 2, and 3 directions, or longitudinal moduli, of a single lamella (Fig. 1b) can be expressed as a function of the individual moduli of the fibers ($f$) and of the extrafibrillar matrix ($m$) (law of mixtures): $C_{11}^{(\text{lamella})} = C_{11}^{(f)} V^{(f)} + C_{11}^{(m)} (1 - V^{(f)})$, $\frac{1}{C_{22}^{(\text{lamella})}} = \frac{V^{(f)}}{C_{22}^{(f)}} + \frac{1 - V^{(f)}}{C_{22}^{(m)}}$, $\frac{1}{C_{33}^{(\text{lamella})}} = \frac{V^{(f)}}{C_{33}^{(f)}} + \frac{1 - V^{(f)}}{C_{33}^{(m)}}$, where $C_{22}^{(m)} = C_{33}^{(m)}$ and $C_{22}^{(f)} = C_{33}^{(f)}$ by symmetry. An individual lamella is thus transverse isotropic with plane of symmetry 2–3 (i.e. orthogonal to the fibril axis).

The corneal stroma is composed of a stack of 300–500 lamellae of varying orientations. The axes of the collagen fibrils in successive lamellae are not truly random, but tend to lie along orthogonal meridians in the medial-lateral and superior-inferior directions, particularly in the posterior cornea[24]. For simplicity, we model the stroma as a stack of layers with half oriented in the medial-lateral direction and half in the superior-inferior direction, which yields:

$$C_{xx}^{(\text{stroma})} = C_{yy}^{(\text{stroma})} = \frac{1}{2} C_{11}^{(\text{lamella})} + \frac{1}{2} C_{33}^{(\text{lamella})} \quad (1a)$$

$$C_{zz}^{(\text{stroma})} = C_{22}^{(\text{lamella})} = C_{33}^{(\text{lamella})} \quad (1b)$$

Here, the $(x, y, z)$ coordinate system is defined in real space such that the $z$-direction is orthogonal to the cornea, and the $x$- and $y$-directions are tangential (medial-lateral and superior-inferior, Fig. 1b). In this model, the cornea is transverse isotropic with plane of symmetry $x$–$y$ (i.e. tangential to the cornea).

For backward Brillouin light scattering spectroscopy with probe light entering the corneal tissue at an angle $\theta$ to the optic axis (Fig. 1c), the Brillouin frequency shift $\Omega$ is given by: $\Omega(\theta) = \frac{2n(\theta)v(\theta)}{\lambda} = \frac{2n(\theta)}{\lambda} \sqrt{\frac{C(\theta)}{\rho}}$, where $n(\theta)$ is the index of refraction of the material, $\lambda$ is the wavelength of light, $v(\theta) = \sqrt{C(\theta)/\rho}$ is the longitudinal-wave acoustic speed, and $C(\theta)$ is the effective longitudinal modulus of the stroma at angle $\theta$ to normal. For simplicity, we consider a case where the incident beam is in the $x$–$z$ plane. The formalism described below can be applied to other tilt directions with appropriate coordinate transformations.

The index of refraction in the cornea is known to differ slightly depending on the direction of light propagation. Mueller matrix

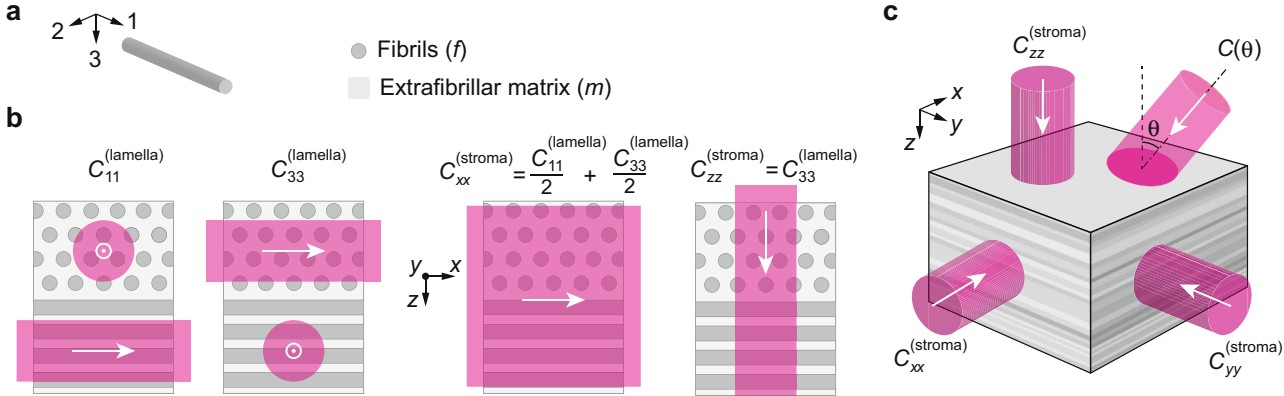

**Fig. 1 Brillouin measurements at different laser incidence angles detect tissue anisotropy. a** Within each lamella, collagen fibers (*f*) are co-aligned within a gel matrix (*m*). We define a coordinate system (1–3) for each lamella in which the 1-direction is always aligned with the fibril axis. **b** The elastic modulus in the 1-direction $C_{11}$ is different than the moduli orthogonal to the fibril axis, $C_{22}$ and $C_{33}$. By reducing the laser spot size, it is possible to resolve individual lamellae using Brillouin microscopy. When the laser spot size covers multiple lamellae, we measure a combined modulus. For the bulk stroma we define the *z*-direction as orthogonal to the cornea and the *x*−*y* plane as tangential. **c** Cut-away view through the stromal depth showing stacked lamellae. The cornea is mechanically anisotropic, meaning that the longitudinal moduli ($C_{ii}$) have directional dependence. This can be probed using Brillouin microscopy with varying laser incidence angle $\theta$.

ellipsometry of the human cornea in vivo found that $n_x - n_z = 1.4 \times 10^{-4}$,[26] similar to other measurements[27]). If individual lamellae are modeled as uniaxial birefringent layers stacked orthogonally, then $n_x - n_z = 2.8 \times 10^{-4}$ theoretically for a single lamella[28]. As we will see, the contribution of this magnitude of birefringence to the Brillouin shift is 2–3 orders of magnitude lower than that of mechanical anisotropy. Therefore, we can neglect the birefringence and assume $n(\theta)$ to be a constant independent of $\theta$.

An exact analytic expression for $C(\theta)$ can be written in terms of the stiffness tensor coefficients (Supplementary Materials). In the case of weak transverse isotropy, i.e. $C_{xx}^{(stroma)} \approx C_{zz}^{(stroma)}$, the effective longitudinal modulus can be expressed (Supplementary Materials) as: $C(\theta) \approx C_{zz} + (C_{xx} - C_{zz})\sin^4\theta + 2(C_{xz} + 2 G_{yz} - C_{zz})\sin^2\theta\cos^2\theta$, where the superscripts were ignored, and $G_{yz}$ ($\ll C_{zz}$) is shear modulus of the stroma in the $y$–$z$ plane. We confirm $C(0) = C_{zz}$, and $C(\pi/2) = C_{xx}$ (the beam is tilted toward the $x$-axis). The difference of these two values characterizes the magnitude of anisotropy of the tissue. We introduce anisotropic parameters, $\alpha_{xx}$ and $\delta$, defined as follows:

$$\alpha_{xx} = C_{xx}/C_{zz} - 1 \qquad (2a)$$

$$\delta = (C_{xz} + 2 G_{yz})/C_{zz} - 1 \qquad (2b)$$

Thus, we can write

$$C(\theta) = C_{zz}\left(1 + \alpha_{xx}\sin^4\theta + 2\delta\sin^2\theta\cos^2\theta\right) \qquad (3)$$

The anisotropic coefficients are analogous, but not identical, to the Thomson parameters used to describe seismic anisotropy (Note: Thomson parameters are defined in terms of wave speeds rather than elastic moduli)[29].

Putting Eqs. (1a) and (1b) into Eqs. (2a) and (2b), we find:

$$\alpha_{xx}^{(stroma)} = \frac{(C_{11}^{(lamella)} + C_{33}^{(lamella)})/2}{C_{33}^{(lamella)}} - 1 = \frac{1}{2}\alpha_{11}^{(lamella)} \qquad (4)$$

Here $\alpha_{11}^{(lamella)}$ corresponds to the anisotropic parameter of a single lamella (with weak anisotropy $C_{33}^{(lamella)} \approx C_{11}^{(lamella)}$). It can be further shown that

$$\alpha_{11}^{(lamella)} = [\beta_2 V^{(f)} + \beta_1(1 - V^{(f)})][\beta_1 V^{(f)} + 1 - V^{(f)}]/\beta_1 - 1, \qquad (5)$$

where $\beta_1 = C_{33}^{(f)}/C_{11}^{(m)}$ and $\beta_2 = C_{33}^{(f)}/C_{11}^{(f)}$ (Supplementary Materials).

**Brillouin imaging of individual corneal lamellae**. To test if Brillouin microscopy can indeed detect the anisotropy of individual lamellae, we excised the cornea from a porcine eye within 4 h of sacrifice. First, the corneal flap was measured en-face and imaged using an inverted confocal Brillouin microscope (Fig. 2a). In this orientation, the laser is orthogonal to almost all collagen fibrils, so we expect the lamellar contrast to disappear. Indeed, the resulting 38 μm × 38 μm Brillouin image is mostly uniform with a Brillouin value of 7.981 ± 0.078 GHz (mean ± standard deviation, measured at 532 nm). Second, we cut the corneal flap across the middle and mounted it vertically, allowing the cross-section to be imaged from beneath using an inverted confocal Brillouin microscope with a lateral resolution of ~0.5 μm (Fig. 2c). Stromal tissue with a cross-sectional area of 30 μm × 30 μm located in the center and the middle third depth-wise was imaged (Fig. 2d). Lamellae in the middle third of the corneal stroma are predominantly orthogonally stacked with little interleaving, resulting in alternating layers of higher and lower elastic modulus (from $C_{11}^{(lamella)}$ to $C_{33}^{(lamella)}$). This pattern is evident in the Brillouin images which show ribbons of width 0.5–5 μm with alternating Brillouin values.

After identifying oriented lamella of lowest and highest Brillouin values (see the "Methods" section), we estimate that $\Omega(0) = 8.032 \pm 0.008$ GHz and $\Omega(\pi/2) = 8.879 \pm 0.018$ GHz. This inter-lamellar contrast of 10.5% in Brillouin values (i.e. $\Omega(\pi/2)/\Omega(0) = 1.105$) corresponds to $\alpha_{xx}^{(lamella)} = C_{xx}/C_{zz} - 1 = (\Omega(\pi/2)/\Omega(0))^2 - 1 = 0.222$. We expect the value of $\Omega(0)$ from the cross-section to approximately match the average value in the en-face image, which also measures $\Omega(0)$. They are indeed similar for this cornea (8.032 and 7.981 GHz, respectively) with the slight difference likely corresponding to spatial variation as these could not be measured at the exact same location.

Figure 2e, f shows TEM images of a single corneal cross-section taken in a similar region (central area of cornea, middle of stromal depth) at two different magnifications. Layers of alternating fibril orientation are visible, as expected from textbook examples[30]. This lamellar structure is consistent with what we observed in the Brillouin images.

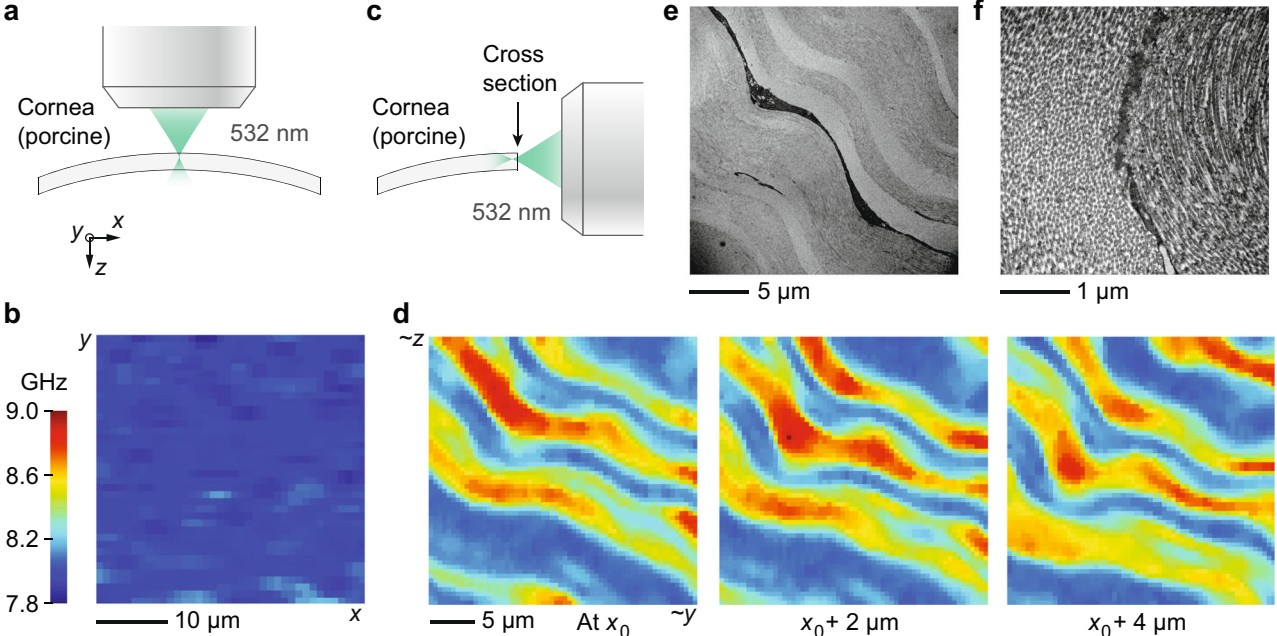

**Fig. 2 High-resolution Brillouin imaging of ex vivo porcine cornea with a 532 nm wavelength. a** Schematic showing en face Brillouin imaging of the cornea using a 532 nm laser focused to a spot size of approximately 0.5 μm. **b** Brillouin image of ex vivo porcine cornea measured en-face, showing relatively uniform Brillouin value across the x–y plane. **c** Schematic showing Brillouin imaging of the corneal cross-section. The laser was focused just below the surface of the cross-section. **d** Brillouin images of corneal cross-section at three different (relative) depths, revealing a lamellar pattern. **e** Transmission electron microscopy (TEM) image of corneal cross-section showing individual lamellae with alternating fibril orientations. A keratocyte is visible in between the two lamellae at the center of the image (keratocyte appears black with substructure). **f** Higher resolution TEM image of the same region. Individual collagen fibrils can be seen aligned to the preferred directions in two adjacent lamellae.

We made further measurements using a confocal Brillouin microscope with a 780 nm laser, observing the same dramatic contrast due to corneal anisotropy. Figure 3a shows the 28 μm × 28 μm en-face image of a (different) cornea, with Brillouin value 5.434 ± 0.012 GHz (mean ± standard deviation, consistent with the wavelength-dependence of the Brillouin shift). Figure 3b shows cross-sectional images of the same cornea at three different (relative) depths. After identifying oriented lamella of lowest and highest Brillouin values as before, we estimate that $\Omega(0) = 5.395 \pm 0.002$ GHz and $\Omega(\pi/2) = 5.916 \pm 0.008$ GHz for this cornea, corresponding to $\alpha_{xx}^{(lamella)} = 0.202$. Again, we expect the value of $\Omega(0)$ from the cross-section to approximately match the average value in the en-face image, which also measures $\Omega(0)$. They are indeed similar for this cornea (5.395 and 5.434 GHz, respectively).

Figure 3c shows a larger (37 μm × 37 μm) cross-sectional Brillouin image of a second cornea (from a different animal). For this sample, we estimate that $\Omega(0) = 5.350 \pm 0.003$ GHz and $\Omega(\pi/2) = 5.798 \pm 0.004$ GHz, corresponding to $\alpha_{xx}^{(lamella)} = 0.174$. Figure 3d, e show Brillouin images of a third cornea (from another pig) measured en-face and in cross-section, respectively, at three different (relative) depths. In the en face images, the Brillouin value remains fairly uniform throughout the depth with an average value of $\Omega(0) = 5.468 \pm 0.007$ GHz. In the cross-sectional images, the lamellar pattern is evident with estimated $\Omega(0) = 5.459 \pm 0.001$ GHz and $\Omega(\pi/2) = 5.988 \pm 0.009$ GHz, corresponding to $\alpha_{xx}^{(lamella)} = 0.203$.

We collected a total of 31 cross-sectional Brillouin images from 7 ex vivo porcine corneas of 7 pigs using the 780 nm-Brillouin microscope. The inter-lamellar difference in the Brillouin frequency varied between corneas, perhaps due to mechanical variation from animal to animal, or variation from location to location in the cornea. Variation may also be attributed to

differences in hydration level (despite our best efforts; see the "Methods" section) and differences in the orientation of the cross-sections or, in other words, the orientation of collagen fibrils with respect to the optical beam. We found that $\Omega(0)$ ranged from 5.310 to 5.461 GHz, and $\Omega(\pi/2)$ ranged from 5.636 to 6.098 GHz. The resulting $\alpha_{xx}^{(lamella)}$ values ranged from 0.126 to 0.247 with mean value 0.184 ± 0.031. The distribution of $\alpha_{xx}^{(lamella)}$ values for all measurements is shown in Fig. 3f.

**Angle-dependence of bulk cornea Brillouin measurements.** The layered structure observed at the microscopic scale translates to a weak anisotropy of the bulk stroma, which we expected to see in larger-scale Brillouin measurements. For measurements of the intact cornea, we used a lower resolution (numerical aperture of ~0.1), near-infrared ($\lambda = 780$ nm) Brillouin imaging system[31]. As before, corneal flaps were excised from porcine eye globes received within 4 h of sacrifice. Intact corneal flaps were mounted in a Barron chamber which could be rotated, allowing for different laser incidence angles at a specific location on the cornea (Fig. 4a). Brillouin measurements of 11 cornea samples were made at 6 different laser incidence angles in either the central cornea or an off-center location.

Results are shown in Fig. 4b (central location) and Fig. 4c (off-center locations). Angles in air were converted to angles in tissue assuming a stromal index of refraction of 1.376[32]. The maximum possible angle is limited by beam aberration or refraction at larger angles. Angle-dependence data were fit to the weak transverse isotropy model (Eq. (3)), with free parameters $\Omega(0)$, $\alpha_{xx}^{(stroma)}$, and $\delta$. Agreement with the model was fairly good, with $R^2$ ranging from 0.74 to 0.96 for the different samples. The fitting results were similar when $\Omega(0)$ was removed as a free parameter. Values and confidence ranges for all fitted parameters are tabulated in the Supplementary Materials, Supplementary Table 1.

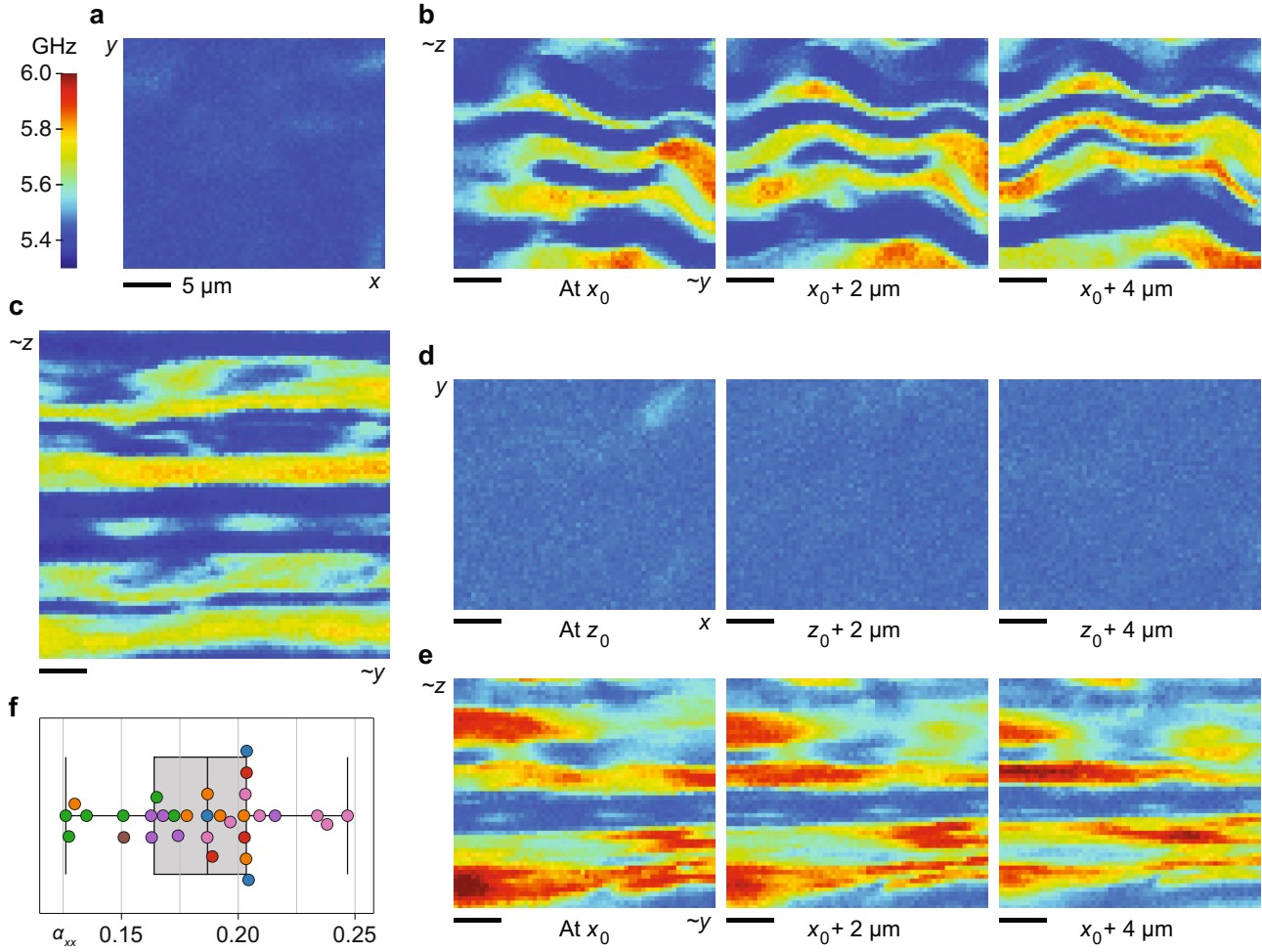

**Fig. 3 High-resolution Brillouin imaging of ex vivo porcine corneas with a 780 nm wavelength. a** En-face image showing relatively uniform Brillouin value across the x–y plane. Note that the colormap scale used in this figure is different from that of Fig. 2b (due to the wavelength difference). **b** Brillouin images of the same corneal sample measured in cross-section at three different (relative) depths, revealing a lamellar pattern. **c** Larger Brillouin image of a second corneal sample measured in cross-section. **d** Brillouin images of a third corneal sample measured en-face at three different (relative) depths. **e** Brillouin images of the same (third) corneal sample measured in cross-section at three different (relative) depths. **f** Distribution of $\alpha_{xx}$ values estimated from 31 cross-sectional Brillouin images of 7 ex vivo corneas from 7 pigs. Points displayed in the same color represent data taken at different spatial locations in the same corneal cross-section. Box-plot: center line, median; box limits, upper and lower quartiles; whiskers, range of data.

Although there was variation from one cornea to another, we did not observe a significant difference in the average fitted anisotropic parameter with $\alpha_{xx}^{(stroma)} = 0.108 \pm 0.020$ in the central cornea, and $\alpha_{xx}^{(stroma)} = 0.110 \pm 0.035$ in other locations. From the cross-sectional imaging experiment described in the previous section, we have measured $\alpha_{xx}^{(lamella)}$ ranging from 0.126 to 0.247 with mean value $0.184 \pm 0.031$. Based on Eq. (4), we expect $\alpha_{xx}^{(stroma)}$ to correspondingly range from ~0.063 to 0.123 with mean value $0.092 \pm 0.015$. This is fairly similar to $\alpha_{xx}^{(stroma)} = 0.109 \pm 0.030$ (average at all locations), as determined from the angle-resolved measurements here.

**Anisotropy in vivo in Brillouin maps of the human cornea.** We also found evidence of corneal anisotropy in Brillouin measurements of human subjects. Study participants with normal corneas ($n = 4$, 3:1 M:F, $31.5 \pm 2.4$ Y/O) were scanned using a custom-built Brillouin imaging system at a wavelength of 780 nm[31]. When a human subject is imaged using this instrument, the subject's gaze angle is directed towards a stationary fixation target while the Brillouin interface is translated right-left and up-down to measure different locations laterally across the cornea. Because

the optical path is rigidly fixed with respect to the interface, as the laser position changes, the angle of incidence (relative to the corneal surface) also changes. When the laser incidence angle is close to 0°, this change is small, but for larger tilt angles (corresponding to a fixation target at ~20°) there is a gradually increasing angle of incidence as the laser moves from right to left across the cornea (see diagrams in Fig. 5a, b).

Data from ~30 axial scans were combined to create a color-coded map of the mean Brillouin shift laterally across the corneal stroma. The irregular shape of the maps is owing to manual positioning of the measurement points using a slit-lamp joystick. An increasing Brillouin gradient from right to left is apparent in the resulting maps, with larger Brillouin values for larger incidence angles as we would expect. The Brillouin maps of three different human subjects displaying this gradient are shown in Fig. 5b. By contrast, when the laser angle is changed to minimize the variation in incidence angle (Fig. 5a), the resulting Brillouin map is more uniform with no obvious gradient.

To analyze the angle-dependence inherent in the human maps, we converted the map coordinates to approximate angles assuming a spherical corneal surface with 7.8 mm radius-of-curvature[30] and fit to the transverse isotropy model (Eq. (3)). The fitted anisotropic

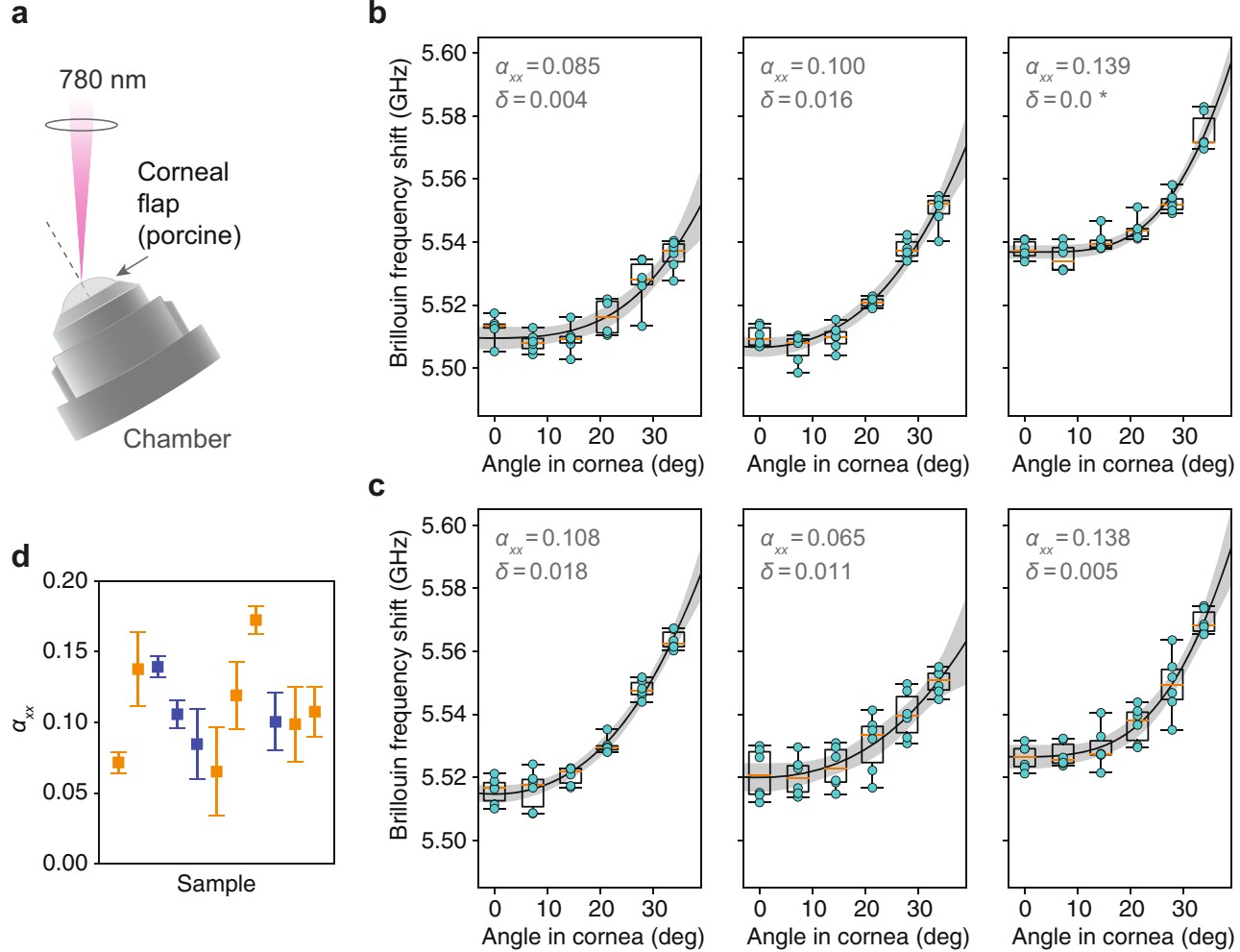

**Fig. 4 Measurement of Brillouin angle-dependence in ex vivo porcine cornea. a** Schematic showing Brillouin angle-dependence measurements of the cornea mounted in a Barron chamber using a 0.1-NA, 780 nm optical beam. **b** Plots of Brillouin frequency shift measured at the center of the cornea with different in-tissue angles for 3 porcine samples (different animals). For each sample, the cycle of angle measurements was repeated six times (alternating increasing and decreasing angle) to minimize the impact of environmental drifts. Individual measurement points are shown in cyan, along with box plots for all six points at each angle setting. Best-fit curves are plotted (solid line) with 95% confidence bands (gray). Fitted values of $\alpha_{xx}$ and $\delta$ are shown. **c** Plots of Brillouin frequency shift measured 2, 3, and 4 mm away from center (left to right, respectively) with different in-tissue angles for three porcine samples (different animals). **d** Distribution of fitted $\alpha_{xx}^{(stroma)}$ values for 11 corneas from 11 different animals. Error bars indicate confidence ranges. Points in blue correspond to the central cornea, while points in orange correspond to other locations 2–4 mm off center. Box-plots: center line, median; box limits, upper and lower quartiles; whiskers, range of data.

parameter values for the human corneas measured with larger tilt angles are $\alpha_{xx}^{(stroma)} = 0.064 \pm 0.065$, $\alpha_{xx}^{(stroma)} = 0.070 \pm 0.012$, $\alpha_{xx}^{(stroma)} = 0.052 \pm 0.010$ (left to right). Agreement with the model was not as good as in the porcine case (previous section), with $R^2$ ranging from 0.46 to 0.60 for these 3 subjects, perhaps because of the relatively crude angle estimation and generally greater variability in the data. For the cornea measured with laser incidence angle close to 0°, $\alpha_{xx}^{(stroma)} = 0.027 \pm 0.021$, but the fit quality was poor ($R^2 = 0.09$), likely due to the small range of angles. Values and confidence ranges for all fitted parameters are tabulated in the Supplementary Materials, Supplementary Table 3.

**Anisotropy in Brillouin maps of the porcine cornea.** Similar to the human measurements in the previous section, we also created Brillouin maps of porcine corneas and analyzed the angle-dependence. Intact porcine eye globes received within 4 hours of sacrifice were scanned using a custom-built Brillouin imaging system at a wavelength of 780 nm[31]. Data from a 7 × 7 grid of

axial scans were combined to create a color-coded map of the mean Brillouin shift laterally across the corneal stroma.

Figure 6a shows the relatively uniform Brillouin map of a porcine cornea scanned with the laser approximately normal to the corneal apex and less variation in incidence angle across the cornea. Figure 6b (leftmost) shows the Brillouin map of the same cornea scanned with the laser tilted by ~15° (with respect to the optical axis) causing greater variation in incidence angle across the cornea, resulting in an increasing Brillouin gradient from right to left. A similar gradient is apparent in the right four Brillouin maps in Fig. 6b, which correspond to the corneas of four other animals also measured with larger tilt angles.

As for the human maps, we converted the map coordinates to angles in order to analyze the Brillouin angle-dependence (Fig. 6c and d). Because the porcine corneas were automatically scanned (unlike the human maps which were taken manually), we were able to use the known coordinates of the corneal surface to compute the individual corneal radii of curvature and more accurately calculate the local incidence angles. The corneal radii

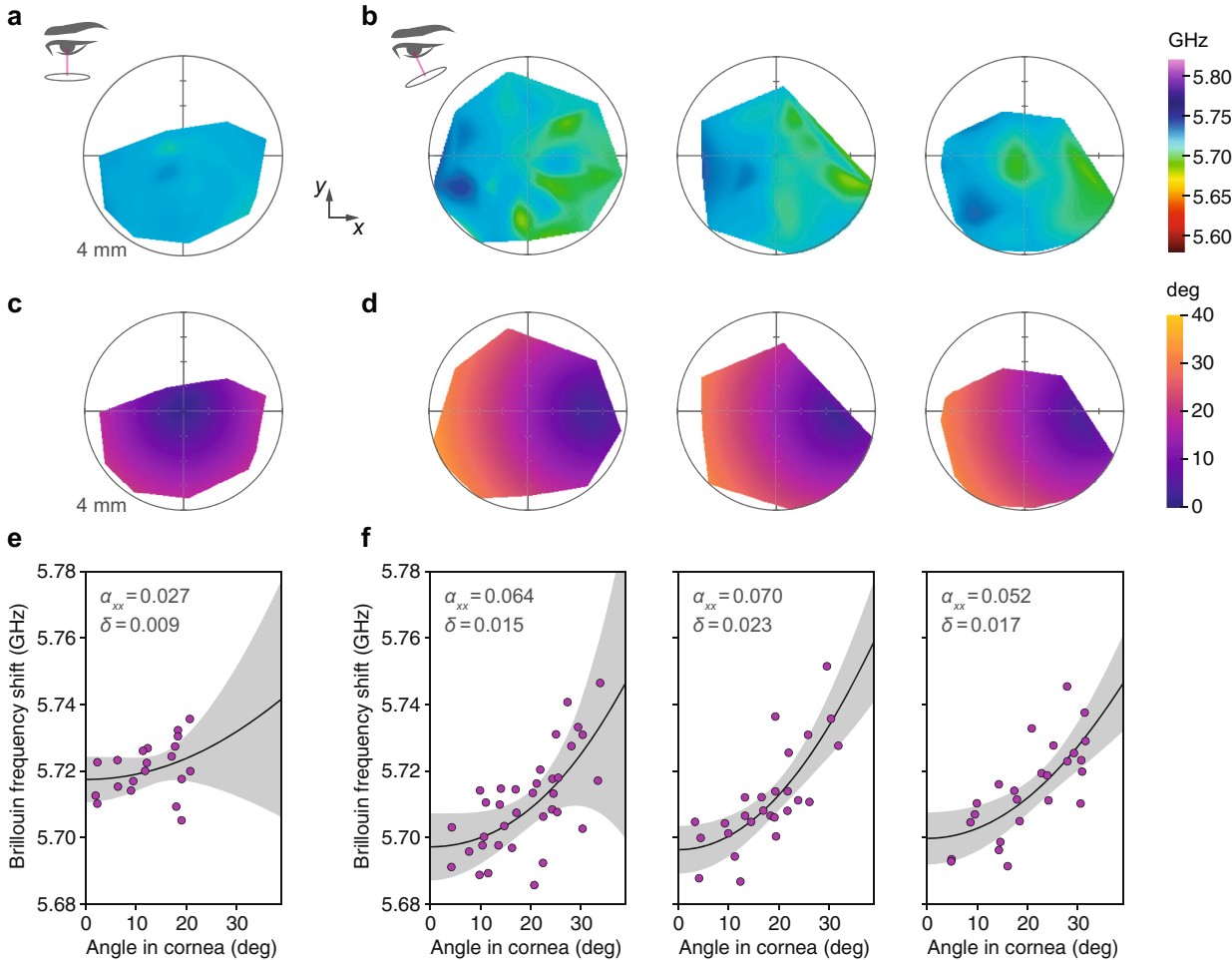

**Fig. 5 Anisotropy in vivo in Brillouin maps of the human cornea ($n = 4$, 3:1 M:F, 31.5 ± 2.4 Y/O). a** Brillouin frequency shift map of a normal subject with laser incidence angle close to 0° at the corneal apex (see diagram at left). **b** Brillouin maps of three normal subjects for a steeper set of laser incidence angles (see diagram). **c** and **d** Maps showing computed laser incidence angle corresponding to the Brillouin maps above. The laser-cornea incidence angle varies slightly as the laser is translated in $x$ (left-right) and $y$ (up-down) to create the Brillouin map due to corneal curvature. **e** and **f** Plot of Brillouin values for each map versus computed incidence angle. The increasing Brillouin shift with angle is consistent with expected corneal anisotropy. Individual map data points (shown in magenta) were fit to the weak transverse isotropy model (Eq. (3)). Best-fit curves are plotted (solid line) with 95% confidence bands (gray). Fitted values of $\alpha_{xx}$ and $\delta$ are shown.

of curvature ranged from 7.61 to 8.79 mm with average 8.33 ± 0.44 mm.

Data were fit to the weak transverse isotropy model (Eq. (3)) as in previous sections. Agreement with the model was fairly good, with $R^2$ ranging from 0.78 to 0.84 for the different samples. Although the angle-dependence data shown in Fig. 6e and in Fig. 6f (leftmost) were measured in the same cornea, there is a discrepancy between the fitted values of $\alpha_{xx}^{(\text{stroma})} = 0.120 \pm 0.010$ and $\alpha_{xx}^{(\text{stroma})} = 0.082 \pm 0.005$ (respectively), perhaps owing to the small range of angles in Fig. 6e. Values and confidence ranges for all fitted parameters are tabulated in the Supplementary Materials, Supplementary Table 2.

For the five corneas shown in Fig. 6f, fitted values of $\alpha_{xx}^{(\text{stroma})}$ ranged from 0.075 to 0.122 with average 0.096 ± 0.018. This value is similar to the fitted value $\alpha_{xx}^{(\text{stroma})} = 0.109 \pm 0.030$ obtained via angle-resolved measurements of the bulk porcine cornea and the estimated value $\alpha_{xx}^{(\text{stroma})} = 0.092 \pm 0.015$ from Brillouin imaging of individual corneal lamellae.

**Longitudinal stiffness coefficients of the cornea.** From the angle-resolved Brillouin measurements of the porcine and human corneas we can estimate the longitudinal moduli. Data were fit to

the weak transverse isotropy model (Eq. (3)), yielding fitted values for $\Omega(0) = \frac{2n}{\lambda}\sqrt{C_{zz}/\rho}$, $\alpha_{xx} = C_{xx}/C_{zz} - 1$, and $\delta = (C_{xz} + 2G_{yz})/C_{zz} - 1$. Assuming a stromal index of refraction[32] of 1.376 and a mass density[33] of 1.05 g cm$^{-2}$, we extracted all three longitudinal moduli $C_{zz}$ and $C_{xx} = C_{yy}$ for each sample as well as the combination of moduli $C_{xz} + 2G_{yz}$ ($\approx C_{xz}$) (Table 1). The symmetry of transverse isotropy dictates that $C_{yz} = C_{xz}$ and $C_{xy} = C_{xx} - 2G_{xy} \approx C_{xx}$. Therefore, all the longitudinal stiffness coefficients $C_{ij}$'s ($i, j = x, y, z$) have been determined.

It is interesting to compare the two species. The human corneas exhibited greater out-of-plane stiffness $C_{zz}$ than the porcine corneas. The in-plane stiffness, $C_{xx}$ and $C_{yy}$, however, was similar between human and porcine corneas. As a result, the human corneas have a lower degree of anisotropy than the porcine corneas.

The anisotropic parameters we determined from measurements of porcine and human corneas can be used in our composite model to estimate the properties of the cornea's constituent parts: collagen fibrils and gel matrix. The gel matrix in particular has not previously been accessible for material characterization. The mean anisotropic parameter measured in humans was $\alpha_{xx}^{(\text{stroma})} = 0.062 \pm 0.008$ (three subjects in Fig. 5f),

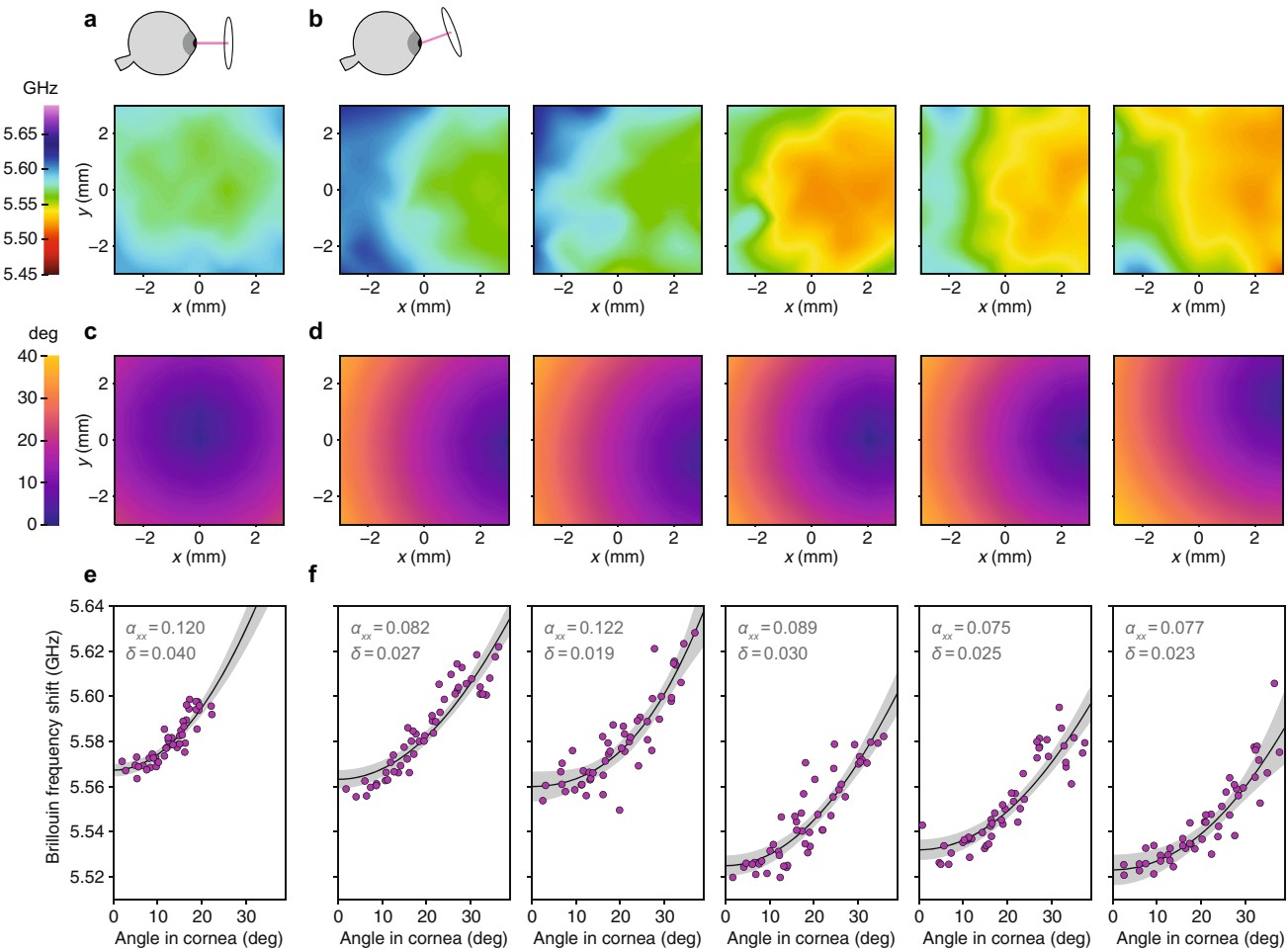

**Fig. 6 Anisotropy in Brillouin maps of ex vivo porcine cornea. a** Brillouin frequency shift map of intact porcine cornea with laser incidence angle close to 0° at the corneal apex (see diagram above). **b** Brillouin map of the same cornea (left) and the corneas of four other animals (right) for a steeper set of laser incidence angles (see diagram above). **c** and **d** Maps showing computed laser incidence angle corresponding to the Brillouin maps above. The laser-cornea incidence angle varies slightly as the laser is translated in x (left-right) and y (up-down) to create the Brillouin map due to corneal curvature. **e** and **f** Plot of Brillouin values for each map versus computed incidence angle. The increasing Brillouin shift with angle is consistent with expected corneal anisotropy. Individual map data points (shown in magenta) were fit to the weak transverse isotropy model (Eq. (3)). Best-fit curves are plotted (solid line) with 95% confidence bands (gray). Fitted values of $\alpha_{xx}$ and $\delta$ are shown.

**Table 1 Longitudinal elastic moduli for porcine corneas ex vivo (at ~ 23°C) and human corneas in vivo measured by angle-resolved Brillouin measurements ($\lambda = 780$ nm).**

| Stiffness coefficients | Porcine ex vivo (GPa) | Human in vivo (GPa) |
|---|---|---|
| $C_{zz}$ | 2.567 ± 0.012 | 2.738 ± 0.001 |
| $C_{xx}$, $C_{yy}$ | 2.848 ± 0.078 | 2.908 ± 0.022 |
| $C_{xz} + 2G_{yz}$ | 2.593 ± 0.024 | 2.789 ± 0.010 |

Mean ± standard deviation from four samples in each group.

corresponding to $\alpha_{xx}^{(lamella)}(human) = 0.124 \pm 0.015$ (Supplementary Materials). For porcine samples, we measured $\alpha_{xx}^{(lamella)}(porcine) = 0.184 \pm 0.031$ (cross-sectional data in Fig. 2f). Considering the different fibril volume fractions[25], $V^{(f)}(human) = 0.22$ and $V^{(f)}(porcine) = 0.28$, and assuming that the material properties of the collagen fibrils and gel matrix are similar between humans and pigs (specifically, $\beta_1$, $\beta_2$ values within 5%), we can estimate the ratio of the longitudinal modulus in the axial '1' direction (along the fiber) to that in the transverse '2,3' directions

(orthogonal to the fiber), $\beta_2(human) = C_{11}^{(f)}/C_{33}^{(f)} = 1.35$ and $\beta_2(porcine) = 1.41$ (Eq. (5)). Similar values have been reported in ultrasound measurements of collagen from the bovine Achilles tendon which found $C_{11}^{(f)}/C_{33}^{(f)} = 1.47$ (fixed)[34], $C_{11}^{(f)}/C_{33}^{(f)} = 1.33$ (fresh)[35]. Likewise, we estimate that the ratio of the longitudinal modulus of the gel matrix (assumed isotropic) to that of the fiber (in the transverse direction) is $\beta_1(human) = C_{11}^{(m)}/C_{33}^{(f)} = 0.70$ and $\beta_1(porcine) = 0.68$, suggesting that the mechanical contrast between matrix and fibers is relatively low in the transverse direction.

## Discussion

We have demonstrated that individual stromal lamellae can be resolved based on their Brillouin shift contrast. This contrast arises directly from the anisotropic elastic properties of the lamellae stemming from their fibrillar organization. We derived a composite model describing the lamellar network of collagen fibrils in the cornea. In aggregate, the crisscrossing lamellae produce a weakly transverse isotropic stroma. We confirmed that the Brillouin frequency shift of the cornea is indeed angle-dependent.

Consistent with our model, the anisotropic parameter $\alpha_{xx}^{(\text{stroma})} = 0.092 \pm 0.015$ estimated from Brillouin imaging of individual cornea lamellae agrees fairly well (within uncertainty ranges) with the values obtained via angle-resolved measurements at a single point, $\alpha_{xx}^{(\text{stroma})} = 0.109 \pm 0.030$, and via angle-resolved analysis of map data, $\alpha_{xx}^{(\text{stroma})} = 0.096 \pm 0.018$. While techniques such as TEM have previously elucidated the fibrillar architecture of the cornea, this work uniquely characterizes the elastic properties of the cornea at the microstructural scale, allowing us to estimate the stiffness coefficients (longitudinal moduli) describing the transverse isotropic tissue. This material information is useful for detailed numerical simulations of the cornea. Equally important, we were able to connect the properties at the micro-scale to those of the bulk cornea, a critical link for understanding the impact of diseases or surgeries that alter the collagen structure.

Fiber-reinforced composite materials are ubiquitous in the body, where their mechanical properties serve their physiological functions. For example, the tympanic membrane of the ear has a lamellar collagen structure similar to that of the cornea and is also mechanically anisotropic[36]. A challenge in reconstructive ear surgery is replicating the complex, anisotropic material properties of the tympanic membrane so that the frequency response and mode shapes required for hearing can be restored[37]. Recent studies have explored laser-induced collagen remodeling as a treatment for chronic inflammatory ear pathologies that aims to restore the intricate collagen network rather than replace it with a poorly matched synthetic material[38]. The microstructure of the uterine cervix also contains an anisotropic network of collagen fibers[39]. Preterm birth is thought to be caused by premature weakening of the cervix under the increasing load of a growing fetus. Efforts are underway to better understand the collagen remodeling process that allows the cervix to become more pliable during gestation in order to devise prophylactics against preterm birth[40]. Likewise, arterial walls have a network of layered collagen and elastin fibers with a predominantly circumferential orientation[41]. Gradual collagen remodeling plays a major role in the weakening and rupture of abdominal aortic aneurysms, resulting in many deaths[42]. Current investigations into the structure-function relationships in soft tissues like these primarily rely on traditional mechanical techniques such as indentation or video extensometry. These techniques cannot probe the micro-scale elastic properties of the tissue, cannot typically probe all axes, and are often difficult to implement in vivo. Multi-angle (vector) Brillouin microscopy may find application in characterizing the anisotropic longitudinal elastic properties of fiber-composite tissues found throughout the body with high spatial resolution.

Another potential avenue of application is biomaterial design. Tissue engineered constructs must be able to closely mimic the in vivo mechanical and structural properties of the tissues they aim to replace. Because many soft tissues exhibit intricate fiber networks with preferential fiber alignment, they typically have complex, anisotropic mechanical properties. This has been recognized and biomaterials that can imitate soft-tissue mechanical anisotropy have been a goal of tissue engineering for more than a decade[43–45]. Brillouin microscopy could provide valuable material data for designing engineered biomaterials and then characterizing the resulting constructs.

Lastly, we observed significant directional dependence in human Brillouin maps measured in vivo. The ~35 MHz gradient over the angle of 0° to 40° is comparable to the magnitude of spatial variation detected in mild corneal disease[31]. Therefore, the angle dependence must be taken into account in the interpretation of raw measurement data. In principle, it can be removed by numerical correction or beam scanning with a constant incident angle.

This work has focused on anisotropy in longitudinal moduli because this is probed by current Brillouin microscopy. However, a complete description of the stiffness or compliance tensor of tissue also requires characterization of shear moduli. In particular, for soft tissues like the cornea, which is nearly incompressible, the shear moduli are used to model corneal deformation under an applied load such as inflation due to intraocular pressure. Unfortunately, at present there are no instruments measuring anisotropy in shear moduli in vivo directly, though developing technologies, such as optical coherence elastography, show promise[46]. Recently, noncontact acoustic microtapping OCE detected large anisotropy in shear moduli in ex vivo porcine corneas[47,48]. Because it measures longitudinal moduli, Brillouin microscopy is particularly sensitive to changes in molecular composition and concentration. In this work, Brillouin contrast arises due to the underlying fiber organization of the cornea. Hence, analyzing the angle-dependence or mechanical anisotropy may be a useful metric to detect disruption or remodeling of the regular collagen lattice due to disease or surgical intervention. We expect the Brillouin angle-dependence to decrease in degenerative conditions where the lamellar or fibrillar microstructure becomes disorganized. It would be interesting to measure the Brillouin angle-dependence in the cone region of a keratoconus cornea.

## Methods

**Cross-sectional Brillouin measurements.** Porcine eye globes were received on ice within 4 h of sacrifice and corneal flaps were immediately excised. A custom 3D-printed fixture was used to mount the corneal flap securely so that a clean slice could be made through the center of the cornea using a scalpel. The resulting slice was then sandwiched between two glass slides so that the cross-section abutted one edge of the slides. The slides and cornea were then mounted vertically in a custom-made holder atop a glass-bottom dish so that the corneal cross-section was optically accessible from below. Empty space in the glass bottom dish and between the slides was filled with corneal preservation medium (Carry-C, Alchimia S.r.l., Padua, Italy) to prevent drying of the corneal slice. The corneal images in Fig. 2 were obtained using a confocal Brillouin microscope with a 532 nm single-frequency laser (Laser Quantum, UK)[49]. The corneal images in Fig. 3 were obtained using a second confocal Brillouin microscope with a 780 nm single-frequency laser (TOPTICA Photonics AG, Germany) and a virtually imaged phased array (VIPA) spectrometer. A co-aligned bright-field microscope was used to navigate to the middle region of the corneal depth for measurement. The Brillouin measurement volume per point was approximately 0.5 µm × 0.5 µm × 1.0 µm and the integration time used was 150–250 ms. Oriented corneal lamellae were identified in the resulting Brillouin image via line-by-line peak/trough detection. Within these regions, the Brillouin values of the upper/lower 10% (respectively) were assumed to correspond to the extremes of fibrillar orientation (in-plane vs. out-of-plane) and averaged to estimate the degree of corneal anisotropy. For TEM analysis, a corneal slice, prepared as above, was fixed using K2 buffer and later imaged using a transmission electron microscope (model CM10, Philips Electron Optics, Eindhoven, The Netherlands).

**Brillouin angle-dependence measurements.** Corneal flaps with a 2–3 mm scleral ring were excised from porcine eye globes using a pair of curved scissors and placed in Carry-C corneal preservation medium (Alchimia S.r.l., Padua, Italy) at room temperature (22–24 °C). To minimize perturbation of the cornea, the iris was not forcibly removed but was instead left attached to the corneal flap. For imaging, the intact corneal flap was mounted in a Barron chamber (Katena Products Inc., New Jersey, USA) filled with Carry-C corneal preservation medium (Alchimia S.r.l., Padua, Italy), which set an effective intraocular pressure (IOP) of about 15 mmHg, confirmed using a standard water column. For these measurements we used a different Brillouin instrument designed for whole-cornea imaging in vivo, at a wavelength of 780 nm. This instrument has been described previously[31] but was upgraded before these measurements to incorporate a Rb vapor cell before the spectrometer to remove elastically scattered light from the Brillouin scattered light for improved spectral extinction. The Barron chamber was mounted on a rotation stage (RP01, Thorlabs Inc., New Jersey, USA) so that the laser incidence angle could be adjusted. A small ink mark was used as a target to ensure that the measurement point remained fixed on the corneal apex as the angle was varied. Because of refraction at the air-cornea interface, we expect the region of tissue being probed at different angles to be slightly offset. Assuming that the properties of the tissue are mostly uniform within a local area, we do not expect this to have a significant effect on the results. The Brillouin measurement volume per point was approximately 2 µm × 2 µm × 30 µm and the integration time used ranged from

200 to 700 ms. The Brillouin value at each angle is obtained by averaging points measured throughout the stromal depth. To minimize the impact of environmental drifts, the cycle of angle measurements was repeated six times (alternating increasing and decreasing angle) and the resulting Brillouin values at each angle were averaged.

**Brillouin maps of human subjects.** Healthy subjects with normal corneas ($n = 4$, 3:1 M:F, 31.5 ± 2.4 Y/O) were imaged at two locations: the Institute for Refractive and Ophthalmic Surgery (IROC) in Zürich, Switzerland, and the Woolfson Eye Institute in Atlanta, Georgia, following approval from the Institutional Review Board (IRB) of Partners HealthCare, the Partners Human Research Committee (PHRC), the Institutional Review Board of IROC, Zürich, and the Woolfson Eye Institute Ethics Committee. Informed consent was obtained from every subject before imaging and all experiments were performed in accordance with the principles of the Declaration of Helsinki. Subjects were not compensated for their participation. In this study, a 'normal cornea' was classified as: <3 diopters refractive error, corneal thickness between 495 and 600 μm, normal corneal topography, no corneal pathology, and no history of eye disease. Study participants were scanned using a clinical Brillouin imaging system at a wavelength of 780 nm which has been described previously[31]. The system specifications are the same as the instrument used for porcine angle-dependence measurements but in a mobile unit. For human subjects, axial scans were taken at ~30 different locations across the cornea with an integration time of 300 ms per point, or 12 s per scan. Brillouin shift values were averaged through the stromal depth, yielding a single Brillouin value per location. A color-coded Brillouin map was then created by 2-dimensional interpolation between these points.

**Brillouin maps of porcine corneas.** Intact porcine eye globes were mounted on an XYZ translation stage (X-XYZ-LSM025A, Zaber Technologies Inc., Vancouver, Canada) for Brillouin measurement with the same instrument used for porcine angle-dependence measurements. Similarly, the Brillouin measurement volume per point was ~2 μm × 2 μm × 30 μm and the integration time used ranged from 200 to 700 ms. The sample was translated 1 mm at a time to create a 7 × 7 grid spanning 6 mm × 6 mm. At each $(x, y)$-coordinate, an axial Brillouin scan was taken and then the corneal surface was moistened with a few drops of Carry-C corneal preservation medium (Alchimia S.r.l., Padua, Italy) in order to prevent drying during the course of measurement. The Brillouin value at each $(x, y)$-coordinate is then obtained by averaging points measured throughout the stromal depth.

**Statistical analysis.** Data analysis was performed using custom software written in Python using the Spyder environment. The SciPy and Lmfit libraries were used for non-linear least-squares minimization and curve-fitting. The R-squared (coefficient of determination) was used as a measure of the global fit of the model and 95% confidence bands are indicated in all plots. Tables of values and confidence intervals for each fitted parameter are included in the Supplementary Materials for all samples/subjects in this study.

**Reporting summary.** Further information on research design is available in the Nature Research Reporting Summary linked to this article.

## Data availability

The processed data that support the findings of this study are included in this published article and its Supplementary Materials. Raw data are available from the corresponding author upon reasonable request.

## Code availability

The custom software used for data acquisition and data analysis are available from the corresponding author upon reasonable request.

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

## Acknowledgements

The authors thank Dr. Theo Seiler and Dr. Doyle Stulting for obtaining the human data and Ms. Neema Kumar and Dr. Jie Zhao for assistance with sample preparation and TEM imaging. This study was supported by funding from the National Institutes of Health, grants R01-EY025454, P41-EB015903, R01-EB027653, and R41-EY028820.

## Author contributions

A.M.E., P.S., and S.-H.Y. designed the experiments. A.M.E. conducted the experiments and analyzed the data. A.M.E. and S.-H.Y. wrote the manuscript.

## Competing interests

A.M.E., P.S., and S.-H.Y. hold issued and pending patents related to the technology. S.-H.Y. is the scientific founder of Intelon Optics, Inc., which licensed the patents.
