## [Peer Review File · Nature Communications]

REVIEWER COMMENTS

Reviewer #1 (Remarks to the Author):

This manuscript demonstrates, for the first time, the anisotropy characterization of ex vivo porcine cornea and in vivo human cornea using Brillouin optical microscopy. In this work, the authors have achieved the following points. (1) They derived a composite mechanical model describing the connection of microstructures such as lamellar networks of collagen fibrils with the bulk behavior of corneal stroma. (2) They measured the mechanical anisotropy of porcine corneas (N = 4) using an angle-resolved equation derived from the composite model, which relates Brillouin shifts per angle with the elastic modulus parameters of a transverse isotropic (TI) model of elasticity. (3) They applied this approach for the measurement of anisotropy of in vivo human corneas (N = 3) using the same angle-resolved equation. The novelty of this work is high.

In terms of significance, the ability to measure anisotropic parameters of the cornea in vivo is nontrivial and has important applications for clinical use such as diagnosis of diseases, monitoring treatments, and conducting a numerical simulation of cornea behavior before surgery for decision-making scenarios. The manuscript might be improved by performing careful statistical validation of the appropriateness of the derived equation and models under the measurement conditions. Specifically, since the calculation of elastic parameters relies on the good fitting of the angle-resolved equation (Eq. 3), careful validation on the confidence range of the fitted parameters, the stability of the results under different initial conditions, and repeatability analysis of the optimized parameters in a larger number of samples is required.

One of the most important assumptions on the derived TI model is that the composite distribution in stroma is the same as described in Figure 1 through the whole xy-plane where the Brillouin imaging has been conducted. Evidence demonstrates that lamella distribution is different among the xy-plane [a], and though the z-axis by having a gradient of Brillouin shift depth-wise [13]. Even though measurements in the porcine cornea were conducted in the same xy-region by tilting the sample with respect to the apex, Brillouin measurements in human corneas were carried out at different sections in the xy-plane per angle. Therefore, the validity of these assumptions needs to be confirmed in ex vivo porcine samples under the same imaging conditions as they were conducted in the in vivo human case.

[a] Meek KM, Knupp C. Corneal structure and transparency. *Prog Retin Eye Res* 49, 1-16 (2015).

Moreover, the impact of the variability of parameters E_{xx} , δ , and $v(0)$ in the curve fitting of Eq. 3 is unknown. In the porcine cornea experiment, E_{xx} is kept constant for all samples during fitting. It is difficult to believe E_{xx} is exactly the same for different cornea samples. Moreover, the utility of fitting Eq. 3 to experimental data is to actually find E_{xx} together with the other parameters, rather than assuming a value for it. On the other hand, $v(0)$ (related to Brillouin shift for 0 degrees) is calculated on the fitting rather than provided by measuring Brillouin shift along 0 degrees. Authors claimed that the SNR is low for 0 degree measurements which avoided its measurement; however, they report it for human experiments in figures Fig. 1a, and 1e, and it has been widely reported in other publications [12,13]. Providing this parameter ($v(0)$) prior to the fitting process can produce more accurate results and prevent overfitting issues. Finally, in Fig. 4f-h, it is difficult to understand

why Exx parameters should range from 0.055 to 0.097 during fitting. What is the impact of the fitting if such a parameter is not bounded in that way?

Some specific questions are:

Is the assumption of lamella distribution along orthogonal meridians true for both corneas in porcine and human cases? Evidence demonstrates differentiated alignment distribution among species.

In the Brillouin imaging of individual corneal lamellae Section, Brillouin images of the corneal cross-section along xy-plane and yz-plane are shown for only one sample. Authors claim that “images taken at different depths were similar” in the xy-plane cross-section, which is opposite to experimental evidence of Brillouin depth-dependent profiles reported before in [12, 13]. Authors need to provide measurements along different stroma depths in order to claim such affirmation. In addition, these measurements need to be conducted for every porcine cornea sample since, as it is clear in Fig. 3b-e, $v(0)$ parameter (intersection of fitting curve with the $x = 0$ axis) is different for each case. Similarly, Brillouin measurements in the yz-plane need to be provided for every sample in order to test the assumption of $Exx = 0.123$ for all porcine corneas.

In Fig. 3, angle-dependent results are shown when the cornea is tilted with respect to the apex; however, in the in vivo human cornea case, different angles are measured at different regions of the cornea taking advantage of the angle between the light beam and the normal of the corneal surface. The validity of this approach needs to be demonstrated in porcine samples as well by (1) generating similar 2D plots, as in Fig. 4, (2) conducting the fitting analysis with Exx also as a variable (not constant) parameter, and (3) validating the fitted parameters Exx and $V(0)$ against the results obtained following Fig 2 analysis for each sample. In this fashion, the appropriateness of the TI model and all the assumptions made during the acquisition will be properly validated and suitable for in vivo human measurements.

In the angle-dependence of bulk cornea Brillouin measurements Section, measurements are only conducted in the same central location of the cornea. What is the variability of the results provided by this method when imaging different regions of the cornea away from the apex?

For all the fittings in this study, R^2 is not enough to qualify the fitting. The authors need to report confidence ranges for every fitting parameter per sample in human and porcine cases.

In the in vivo human Brillouin Section, the tilted imaging version will allow detecting larger gradients for larger angles of incidence. This is only true by assuming that cornea has a cylindrical shape, which is why the authors average Brillouin measurements maps in Figs. 4b-d along the $y=0$ axis (2 mm section along y-axis). This approach may produce less accurate results (averaging in a large 2 mm window), and disregard important data in other regions of the map. The following computer-generated maps in Fig.R1 shows the variation of Theta (color-coded), defined as the absolute value of the angle between the imaging axis and the normal vector of the surface of the cornea (modeled as a sphere), when the imaging axis is parallel to z-axis and tilted certain degree. Darker blue color approximates to $\Theta = 0$ while orange color approximate to larger Θ angles. Fig. R1b approximates to cases in Fig. 4b-d, while Fig. R1a to Fig. 4a. We strongly recommend authors to conduct the correct transformation of angles taking into account the curvature of the cornea along the xy-plane (instead of just $y = 0$) when plotting Brillouin parameters to increase the accuracy of estimations.

Figure R1: Color-coded variation of Theta, defined as the absolute value of the angle between the imaging axis and the normal vector of the surface of the cornea (modeled as a sphere), when the imaging axis is parallel to z-axis (a) and tilted certain degree (b).

What is the tilting angle used in the in vivo measurements?

Is there any impact in the variability of Brillouin shift calculation when imaging the same sample with at two different wavelengths for the macro-scale and micro-scale cases?

It would be very interesting to explore anisotropy estimation in porcine corneas using different levels of the intra ocular pressure (IOP). Experimental evidence in [b, 22] in the elastography of porcine cornea suggests that the level of anisotropy increases with the IOP. It would be important to comment on this topic in the Discussion Section.

[b] Zvietcovich F, Pongchalee P, Meemon P, Rolland JP, Parker KJ. Reverberant 3D optical coherence elastography maps the elasticity of individual corneal layers. Nature Communications 10, 4895 (2019).

What are the main limitations of this approach overall?

The cropped areas in the 2D maps of Fig. 4 were removed due to low SNR. The authors need to add this information during the explanation of results.

Why the in vivo human data plots in Fig. 4e-h are less variable than ex vivo plots in Fig. 3b-e? It is expected to have less variability in ex vivo samples since they are better control conditions. Please, explain.

Supplementary material:

Why the shear terms G_{yz} , G_{xz} were assumed equal (as in a transverse isotropic case)? So far, all the derivations were made for C_{ij} terms for $j,j=[1,2,3]$, and not shear terms.

Equation 12 is missing a C_{zz} term multiplying the whole equation.

There is no reference from the prior work cited for the affirmation that “anisotropy of the cornea is also relatively small”.

Minor issues:

Figure 1: Awkward labeling order.

Figure 4: scale in the 2D plots needs to be provided.

Page 8, line 3: Fig. 4 is too general to describe the angle effect. Please use which figures (a, b, or c, etc) you are referring to.

What is the bibliographic reference for assuming 7.7 mm radius-of-curvature in the human cornea?

In the discussion section, when reporting “Results agree fairly well with our...”, the authors need to define quantitatively at what degree results agree.

Measuring mechanical anisotropy of the cornea with Brillouin microscopy

AMIRA M. ELTONY, PENG SHAO, AND SEOK-HYUN YUN

General Comments

This is a generally well-constructed paper describing Brillouin microscopy in the cornea to quantify mechanical anisotropy. The cornea provides the primary refractive power in the human visual system and its shape is directly linked to its performance as an optical lens. Its elastic properties, therefore, help determine overall visual performance. Any changes in these properties with pathology are of great clinical significance.

Brillouin microscopy probes the longitudinal mechanical properties of a specimen at optical resolution. This technology has been applied to the cornea, but there have been no definitive studies of mechanical anisotropy using Brillouin tools. This article appears to be the first addressing this point.

Characterizing corneal anisotropy is important in predicting its response to different mechanical stimuli. Previous studies on anisotropy using other non-invasive tools such as ultrasound supersonic shear wave imaging and optical coherence elastography have mostly explored tangential anisotropy. Given its lamellar structure, it's clear that tangential is not the primary form of anisotropy in the cornea.

The authors show that the longitudinal elastic moduli of the cornea can be accurately represented as transversely isotropic. This mechanical model more closely matches tissue architecture and makes sense as the next level of complexity compared to a simple isotropic one. Indeed, the primary contribution of this paper is to show that a physically motivated TI model accurately describes measured anisotropy in longitudinal mechanical moduli. However, these measurements do not address the primary feature of a TI model. Namely, it also is needed to describe the anisotropy in **shear** moduli.

Corneal shape is primarily determined by its response to the mechanical load induced by intraocular pressure. The appropriate modulus describing deformation under tensile or inflation loading is the Young's modulus. In a nearly incompressible isotropic material, such as most soft tissue, the Young's modulus is simply related to the shear modulus.

In a TI model, there are two shear moduli. The Young's modulus for a nearly incompressible TI material, such as the cornea, is primarily determined by one of the shear moduli and its response to torsion is primarily determined by the other shear modulus (see specific comment below). Consequently, measurements of anisotropic longitudinal elastic moduli cannot be used to predict how the cornea will deform under mechanical loading but they can infer some underlying properties of the fiber matrix creating a TI structure.

This paper highlights a fundamental issue with Brillouin microscopy for mechanical modeling of the cornea. There is no direct link between longitudinal and shear elastic moduli, especially in nearly incompressible materials such as soft tissue. Since the deformational properties of soft tissue are determined primarily by shear moduli, it is not clear at all if measurements of anisotropic longitudinal moduli will have any clinical relevance. The authors must address this point to properly judge the value of their contribution.

Specific Comments

Abstract, last sentence – The potential for clinical applications remains unclear since only longitudinal moduli are measured. This sentence should be qualified.

Introduction, end of last paragraph – There has been long-term interest in understanding the deformation of soft tissue under mechanical loading. However, for a nearly incompressible material like soft tissue, deformational properties are almost exclusively determined by shear moduli, which are not the focus of this study.

Introduction, 2nd paragraph, 3rd sentence – The Brillouin frequency shift is related to the longitudinal elastic properties of the material. This should be explicitly noted.

Results, Composite model of corneal anisotropy – This model is well argued and makes physical sense in describing the expected longitudinal mechanical properties of the cornea.

Pg. 4, Eqs. 4 and 5 – It's unfortunate that the authors chose the symbol ε to characterize the anisotropy parameter. It may be confused with the strain since ε is often used as the symbol for strain in continuum mechanics. The authors should consider changing this notation to make it very clear that these equations are the anisotropy parameter.

Figure 2 – These results clearly show corneal anisotropy and the lamellar structure. They strongly support a TI model.

Figure 3, minor point – The angle in tissue is compensated for refraction at the air-cornea interface, but is the same tissue probed at different angles? Doesn't the beam probe different sections of the cornea for different angles due to refraction?

Figure 4, minor point – Again, the angle is changed with scanning, but isn't a different section of the cornea probed at each angle? Does this affect measurement variability?

General comment on Figures 3 and 4 – How are Brillouin measurements in soft tissue validated? Is there a gold standard with known longitudinal anisotropy that can be used to compare to corneal measurements to validate the findings?

Table 1 – The findings in this table are generally consistent with ultrasound speed measurements probing longitudinal moduli in soft tissue.

Discussion, 1st paragraph, last two sentences – These conclusions are speculative. Detailed numerical simulations of the cornea focus on tissue deformations under applied loads. Such deformations are primarily determined by the Young's and shear moduli. In the limit of an incompressible material, both the Young's modulus and shear modulus are derived from the material parameters labeled G_{yz} and G_{xy} . That is, the primary moduli describing deformation of the cornea under applied load (e.g, intraocular pressure) cannot be directly identified from the longitudinal moduli. In particular, the Young's modulus is approximately $3G_{xy}$ and is not captured at all by the Brillouin measurements reported here. The authors need to greatly justify the statement that Brillouin measurements can affect clinical understanding when they cannot be directly used to predict the most important feature of mechanical modeling.

Discussion, 2nd paragraph, last sentence – Again, needs to be justified, similar to comments on the 1st paragraph in this section.

Discussion, 3rd paragraph, last sentence – Again, needs to be justified, similar to comments on the 1st paragraph in this section.

Discussion, 4th paragraph, last sentence – Again, needs to be justified, similar to comments on the 1st paragraph in this section.

The discussion section focuses on the potential applications of Brillouin microscopy to predict the biomechanical behavior of many types of soft tissue. This has to be justified given that deformations are primarily determined by shear characteristics and Brillouin microscopy probes longitudinal characteristics.

Methods, Cross-sectional Brillouin measurements – These measurements have high spatial resolution, as noted, but take a very long time to perform. How practical will they be for clinical applications?

Methods, Brillouin maps of human subjects – These measurements are not high spatial resolution and still take over 10 seconds. How practical can they be for general clinical use?

Supplementary Materials – Eq. 11 – See note above about notation. The authors should consider changing this notation to make it very clear that these equations are the anisotropy parameter.

POINT BY POINT RESPONSE TO REVIEWER COMMENTS

We thank both reviewers for their detailed feedback and constructive suggestions for improvement. Based on this input, we have upgraded our system and acquired a significant volume of additional data, which we believe has greatly improved the manuscript. We were delayed in submitting this revision due to several factors: (1) Restricted lab access due to the COVID-19 pandemic, (2) Necessary upgrades to our instrument to enable Brillouin measurement at a 0° incidence angle (with long component lead-times due to global supply chain issues), (3) Limited access to tissue samples due to interrupted service during the COVID-19 pandemic.

Reviewer #1 (Remarks to the Author):

This manuscript demonstrates, for the first time, the anisotropy characterization of ex vivo porcine cornea and in vivo human cornea using Brillouin optical microscopy. In this work, the authors have achieved the following points. (1) They derived a composite mechanical model describing the connection of microstructures such as lamellar networks of collagen fibrils with the bulk behavior of corneal stroma. (2) They measured the mechanical anisotropy of porcine corneas (N = 4) using an angle-resolved equation derived from the composite model, which relates Brillouin shifts per angle with the elastic modulus parameters of a transverse isotropic (TI) model of elasticity. (3) They applied this approach for the measurement of anisotropy of in vivo human corneas (N = 3) using the same angle-resolved equation. The novelty of this work is high.

Response 1: Thanks for the comment on the achievements and novelty of this work.

In terms of significance, the ability to measure anisotropic parameters of the cornea in vivo is nontrivial and has important applications for clinical use such as diagnosis of diseases, monitoring treatments, and conducting a numerical simulation of cornea behavior before surgery for decision-making scenarios. The manuscript might be improved by performing careful statistical validation of the appropriateness of the derived equation and models under the measurement conditions. Specifically, since the calculation of elastic parameters relies on the good fitting of the angle-resolved equation (Eq. 3), careful validation on the confidence range of the fitted parameters, the stability of the results under different initial conditions, and repeatability analysis of the optimized parameters in a larger number of samples is required.

Response 2: Thanks for the suggestion for improvement. Along with increasing the sample sizes, we have reassessed the statistical validity of our analysis and have revised the manuscript accordingly.

Changes made: Angle-dependence data for all samples/subjects were independently fit to the weak transverse isotropy model (Eq. 3) with free parameters $\Omega(0)$, $\alpha^{(\text{stroma})}_{xx}$, and δ (i.e., no assumptions are made any longer about their values). We found that the fitting results were not significantly different when $\Omega(0)$ was removed as a free parameter. We added tables of fitted values and confidence intervals for every fitted parameter for all samples/subjects in this study, along with the R-squared (coefficient of determination), in the Supplementary Materials (Tables S1, S2, S3). 95% confidence bands have also been indicated in all plots (Fig. 4, 5, 6).

One of the most important assumptions on the derived TI model is that the composite distribution in stroma is the same as described in Figure 1 through the whole xy-plane where the Brillouin imaging has been conducted. Evidence demonstrates that lamella distribution is different among the xy-plane [a], and though the z-axis by having a gradient of Brillouin shift depth-wise [13]. Even though measurements in the porcine cornea were conducted in the same xy-region by tilting the sample with respect to the apex, Brillouin measurements in human corneas were carried out at different sections in the xy-plane per angle. Therefore, the validity of these assumptions needs to be confirmed in ex vivo porcine samples under the same imaging conditions as they were conducted in the in vivo human case.

[a] Meek KM, Knupp C. Corneal structure and transparency. Prog Retin Eye Res 49, 1-16 (2015).

Response 3: Thanks for this insightful comment. Our model assumed global transverse isotropy of the stroma, but we agree that the actual cornea deviates from perfect lateral symmetry, and it is worthwhile to investigate this. We performed additional experiments to measure the Brillouin angle-dependence at different locations in porcine corneas and described our findings in the revised manuscript. Although there was some variation from one cornea to another, we did not observe a significant difference in the average fitted anisotropic parameter in the central cornea vs. in other locations. We also created Brillouin maps of porcine corneas, similar to the human maps, and included this new data in the revised manuscript. Analyzing the angle-dependence from these porcine maps to estimate $\alpha^{(\text{stroma})}_{xx}$, we found good agreement with the anisotropy parameter ($\alpha^{(\text{stroma})}_{xx}$) extracted from porcine measurements at a single-point.

Changes made: Fig. 4 has been updated with new data, along with the corresponding Results section, “Angle-dependence of bulk cornea Brillouin measurements”. We have also added a new figure (Fig. 6) and a new section under Results, “Anisotropy in Brillouin maps of the porcine cornea”, which present 2D Brillouin maps of porcine corneas (similar to the human maps).

Moreover, the impact of the variability of parameters Exx, delta, and $v(0)$ in the curve fitting of Eq. 3 is unknown. In the porcine cornea experiment, Exx is kept constant for all samples during fitting. It is difficult to believe Exx is exactly the same for different cornea samples. Moreover, the utility of fitting Eq. 3 to experimental data is to actually find Exx together with the other parameters, rather than assuming a value for it. On the other hand, $v(0)$ (related to Brillouin shift for 0 degrees) is calculated on the fitting rather than provided by measuring Brillouin shift along 0 degrees. Authors claimed that the SNR is low for 0 degree measurements which avoided its measurement; however, they report it for human experiments in figures Fig. 1a, and 1e, and it has been widely reported in other publications [12,13]. Providing this parameter ($v(0)$) prior to the fitting process can produce more accurate results and prevent overfitting issues. Finally, in Fig. 4f-h, it is difficult to understand why Exx parameters should range from 0.055 to 0.097 during fitting. What is the impact of the fitting if such a parameter is not bounded in that way?

Response 4: We appreciate this comment. It is quite challenging to measure the Brillouin shift with a beam incidence angle of 0°, particularly in an optical system with low-NA objective lenses (meaning coarser optical sectioning) as was used here, because of the strong specular reflection from the corneal surface which dwarfs the Brillouin signal and saturates the spectrometer, even well into the corneal depth. In the human data, we were not actually able to make measurements at a 0° incidence angle but have interpolated between nearby points made at small angles to create the continuous Brillouin maps shown in Fig. 5. Over the past several months during this revision, we made considerable efforts to upgrade the laser source and spectral analysis set-up, making measurement at 0° incidence angle possible! Using the upgraded system, we conducted the porcine cornea experiment again with 11 different samples and have described the results in the revised manuscript.

Changes made: New data (including measurements at 0° incidence angle) are shown in Fig. 4. In the revised manuscript, we no longer assume that $\alpha^{(\text{stroma})}_{xx}$ is similar for all porcine corneas. Please see Response 2 (Changes made) above where we discuss how we improved the fitting to the model.

Some specific questions are:

Is the assumption of lamella distribution along orthogonal meridians true for both corneas in porcine and human cases? Evidence demonstrates differentiated alignment distribution among species.

Response 5: To our knowledge, there is a gross similarity in the stromal architecture between the human and porcine corneas [1]. We took several cross-sectional images of the porcine cornea using TEM (examples are shown in Fig. 2E-F) and observed a similar lamellar pattern as has been observed in human samples [2].

[1] Meek, K. M. & Leonard, D. W. Ultrastructure of the corneal stroma: a comparative study. *Biophys. J.* 64, 273–280 (1993).

[2] Komai, Y. & Ushiki, T. The three-dimensional organisation of collagen fibrils in the human cornea and sclera. *Investig. Ophthalmol. Vis. Sci.* 32, 2244–2258 (1991).

In the Brillouin imaging of individual corneal lamellae Section, Brillouin images of the corneal cross-section along xy-plane and yz-plane are shown for only one sample. Authors claim that “images taken at different depths were similar” in the xy-plane cross-section, which is opposite to experimental evidence of Brillouin depth-dependent profiles reported before in [12, 13]. Authors need to provide measurements along different stroma depths in order to claim such affirmation. In addition, these measurements need to be conducted for every porcine cornea sample since, as it is clear in Fig. 3b-e, $v(0)$ parameter (intersection of fitting curve with the $x = 0$ axis) is different for each case. Similarly, Brillouin measurements in the yz-plane need to be provided for every sample in order to test the assumption of $E_{xx} = 0.123$ for all porcine corneas.

Response 6: We performed additional experiments and described our findings in the revised manuscript. Briefly, there is substantial sample-to-sample variability, but within the same cornea, the depth dependent variation in anisotropy was statistically insignificant.

Changes made: New data are shown in Fig. 3. In particular, Fig. 3D and 3E show the depth dependence en-face and in cross-section (respectively) for the same cornea. In the revised manuscript, we no longer assume that $\alpha^{(\text{stroma})}_{xx}$ is similar for all porcine corneas. Please see Response 2 (Changes made) above where we discuss how we improved the fitting to the model.

In Fig. 3, angle-dependent results are shown when the cornea is tilted with respect to the apex; however, in the in vivo human cornea case, different angles are measured at different regions of the cornea taking advantage of the angle between the light beam and the normal of the corneal surface. The validity of this approach needs to be demonstrated in porcine samples as well by (1) generating similar 2D plots, as in Fig. 4, (2) conducting the fitting analysis with E_{xx} also as a variable (not constant) parameter, and (3) validating the fitted parameters E_{xx} and $V(0)$ against the results obtained following Fig 2 analysis for each sample. In this fashion, the appropriateness of the TI model and all the assumptions made during the acquisition will be properly validated and suitable for in vivo human measurements.

Response 7: We have performed the suggested experiments to validate the method and described our findings in the revised manuscript.

Changes made: We have added a new figure (Fig. 6) and a new section under Results, “Anisotropy in Brillouin maps of the porcine cornea”, which present 2D Brillouin maps of porcine corneas (similar to the human maps). Analyzing the angle-dependence from these porcine maps to estimate $\alpha^{(\text{stroma})}_{xx}$, we found good agreement with the anisotropy parameter ($\alpha^{(\text{stroma})}_{xx}$) extracted from porcine measurements at a single-point.

In the angle-dependence of bulk cornea Brillouin measurements Section, measurements are only conducted in the same central location of the cornea. What is the variability of the results provided by this method when imaging different regions of the cornea away from the apex?

Response 8: We have performed additional experiments, taking advantage of improved measurement capability (see Response 4). We measured the angle-dependence in porcine corneas both in the center and away from the center and described our findings in the revised manuscript. Although there was some variation from one cornea to another, we did not observe a significant difference in the average fitted anisotropic parameter in the central cornea vs. in other locations.

Changes made: Fig. 4 has been updated with new data, along with the corresponding Results section, “Angle-dependence of bulk cornea Brillouin measurements”.

For all the fittings in this study, R^2 is not enough to qualify the fitting. The authors need to report confidence ranges for every fitting parameter per sample in human and porcine cases.

Response 9: Thanks for the suggestion. We have done this.

Changes made: We added tables of fitted values and confidence intervals for every fitted parameter for all samples/subjects in this study, along with the R-squared (coefficient of determination), in the Supplementary Materials (Tables S1, S2, S3). 95% confidence bands have also been indicated in all plots (Fig. 4, 5, 6).

In the *in vivo* human Brillouin Section, the tilted imaging version will allow detecting larger gradients for larger angles of incidence. This is only true by assuming that cornea has a cylindrical shape, which is why the authors average Brillouin measurements maps in Figs. 4b-d along the $y=0$ axis (2 mm section along y -axis). This approach may produce less accurate results (averaging in a large 2 mm window), and disregard important data in other regions of the map. The following computer-generated maps in Fig.R1 shows the variation of Theta (color-coded), defined as the absolute value of the angle between the imaging axis and the normal vector of the surface of the cornea (modeled as a sphere), when the imaging axis is parallel to z -axis and tilted certain degree. Darker blue color approximates to Theta = 0 while orange color approximate to larger Theta angles. Fig. R1b approximates to cases in Fig. 4b-d, while Fig. R1a to Fig. 4a. We strongly recommend authors to conduct the correct transformation of angles taking into account the curvature of the cornea along the xy -plane (instead of just $y = 0$) when plotting Brillouin parameters to increase the accuracy of estimations.

Figure R1: Color-coded variation of Theta, defined as the absolute value of the angle between the imaging axis and the normal vector of the surface of the cornea modeled as a sphere), when the imaging axis is parallel to z -axis (a) and tilted certain degree (b).

Response 10: Our original analysis indeed made this simplification—neglecting the curvature in the vertical direction in the region near to the equator—which we thought was a reasonable assumption given that the effect of the curvature is small compared to the noise and measurement uncertainty. Additionally, because of the eyelids, our Brillouin maps of humans are typically wider than they are tall. Nonetheless, we reanalyzed the human maps (and the newly added porcine maps), taking into account the full spherical curvature, and have updated the manuscript accordingly.

Changes made: Fig. 5 has been updated accordingly. Instead of vertical averaging, the plots now show all of the individual Brillouin measurement points of each map (i.e. the points that are interpolated between to create the continuous map) plotted versus incidence angle, taking into consideration the full spherical shape of the cornea. The new Fig. 6 also includes the same analysis.

What is the tilting angle used in the *in vivo* measurements?

Response 11: The tilt angle was $\sim 20^\circ$.

Changes made: This information has been noted in Results, section “Anisotropy *in vivo* in Brillouin maps of the human cornea”.

Is there any impact in the variability of Brillouin shift calculation when imaging the same sample with at two different wavelengths for the macro-scale and micro-scale cases?

Response 12: The wavelength dependence of the Brillouin frequency shift is well established, $\Omega = 2nv/\lambda$. The difference in refractive index should be negligible. Nonetheless, because so much additional data was added as part of this revision, we were able augment the single dataset that was taken at a laser wavelength of 532 nm (the

microscopic data) with a larger volume of data taken using a newer (similar) microscope with a laser wavelength of 780 nm. Results measured at 532 nm and 780 nm are consistent as expected.

Changes made: We have added additional microscope data acquired at a wavelength of 780 nm in Figure 3.

It would be very interesting to explore anisotropy estimation in porcine corneas using different levels of the intra ocular pressure (IOP). Experimental evidence in [b, 22] in the elastography of porcine cornea suggests that the level of anisotropy increases with the IOP. It would be important to comment on this topic in the Discussion Section. [b] Zvietcovich F, Pongchalee P, Meemon P, Rolland JP, Parker KJ. Reverberant 3D optical coherence elastography maps the elasticity of individual corneal layers. Nature Communications 10, 4895 (2019).

Response 13: Thanks for this suggestion. We agree that this would be a very interesting investigation, but wish to consider it for future study as it is beyond the scope of this manuscript.

What are the main limitations of this approach overall?

Response 14: We have added discussion on the limitations of this approach in the revised manuscript.

Changes made: A paragraph was added at the end of the Discussion section to clarify this.

The cropped areas in the 2D maps of Fig. 4 were removed due to low SNR. The authors need to add this information during the explanation of results.

Response 15: This is not entirely correct. Because we use a slit-lamp interface with manual positioning for human measurement, we select the measurement points by hand one-by-one. This is what gives the human maps their irregular boundaries. Depending on the time available with the patient for scanning, we may be able to measure points covering a smaller or a larger area. It is true that we sometimes need to remove the odd point due to low SNR (for example if the subject blinked or moved their head during an axial scan) but this doesn't typically affect the overall shape of the map.

Changes made: An explanation about this has been added in Results, section "Anisotropy *in vivo* in Brillouin maps of the human cornea".

Why the *in vivo* human data plots in Fig. 4e-h are less variable than *ex vivo* plots in Fig. 3b-e? It is expected to have less variability in *ex vivo* samples since they are better control conditions. Please, explain.

Response 16: In the original human data plots, we had averaged vertically over points within 1 mm of the equator, which meant that the point-to-point variability was reduced (through averaging). However, as discussed in Response 10, we have updated Fig. 5 to show the individual data points making up the full map vs. angle (i.e. no averaging). With both figures now showing the same thing, if we compare the updated Fig. 5 to Fig. 4, we can see that the human data plots are indeed more variable than the *ex vivo* plots, as expected.

Supplementary material:

Why the shear terms G_{yz} , G_{xz} were assumed equal (as in a transverse isotropic case)? So far, all the derivations were made for C_{ij} terms for $j,j=[1,2,3]$, and not shear terms.

Response 17: We assume this comment is referring to Eq. S9, which represents the stress-strain relation of a transverse-isotropic material, in which G_{yz} , G_{xz} are indeed equal. Our derivations focus on the C_{ij} terms because the Brillouin frequency shift that we measure is related to the longitudinal moduli (C_{ij} terms) not the shear moduli (G_{ij} terms).

Equation 12 is missing a Czz term multiplying the whole equation.

Response 18: Good eye! Thanks. We have corrected this in the revised manuscript.

Changes made: Eq. S12 was corrected in the Supplementary Materials.

There is no reference from the prior work cited for the affirmation that “anisotropy of the cornea is also relatively small”.

Response 19: This should have been stated as an assumption rather than a fact. However, this assumption was borne out in the Brillouin data presented in the main text.

Changes made: This has been reworded to say, “Assuming that the anisotropy of the cornea is also relatively small”.

Minor issues:

Figure 1: Awkward labeling order.

Response 20: The order was chosen to logically connect the diagrams to corresponding data. But we will defer to the opinion of the Production Editor.

Figure 4: scale in the 2D plots needs to be provided.

Response 21: Done.

Changes made: The scale has been added in Fig. 5.

Page 8, line 3: Fig. 4 is too general to describe the angle effect. Please use which figures (a, b, or c, etc) you are referring to.

Response 22: Done.

Changes made: Changed text from “(Fig. 5)” to “(see diagrams in Fig. 5A-B)” in revised manuscript.

What is the bibliographic reference for assuming 7.8 mm radius-of-curvature in the human cornea?

Response 23: Many references have described the ~7.8 mm radius-of-curvature in the human cornea, e.g. [1].
[1] Dawson, D. G., Ubels, J. L. & Edelhauser, H. F. Cornea and Sclera. In Adler’s Physiology of the Eye, 71–130 (Elsevier Inc., 2011), eleventh edn.

Changes made: We added a reference for this (Ref. 32) in the revised manuscript.

In the discussion section, when reporting “Results agree fairly well with our...”, the authors need to define quantitatively at what degree results agree.

Response 24: Thanks. We have revised the manuscript accordingly.

Changes made: A quantitative comparison has been added in the first paragraph of the Discussion section.

Reviewer #2 (Remarks to the Author):

This is a generally well-constructed paper describing Brillouin microscopy in the cornea to quantify mechanical anisotropy. The cornea provides the primary refractive power in the human visual system and its shape is directly linked to its performance as an optical lens. Its elastic properties, therefore, help determine overall visual performance. Any changes in these properties with pathology are of great clinical significance. Brillouin microscopy probes the longitudinal mechanical properties of a specimen at optical resolution. This technology has been applied to the cornea, but there have been no definitive studies of mechanical anisotropy using Brillouin tools. This article appears to be the first addressing this point. Characterizing corneal anisotropy is important in predicting its response to different mechanical stimuli. Previous studies on anisotropy using other non-invasive tools such as ultrasound supersonic shear wave imaging and optical coherence elastography have mostly explored tangential anisotropy. Given its lamellar structure, it's clear that tangential is not the primary form of anisotropy in the cornea.

Response 25: Thanks for the encouraging comments and pointing out the potential clinical impact.

The authors show that the longitudinal elastic moduli of the cornea can be accurately represented as transversely isotropic. This mechanical model more closely matches tissue architecture and makes sense as the next level of complexity compared to a simple isotropic one. Indeed, the primary contribution of this paper is to show that a physically motivated TI model accurately describes measured anisotropy in longitudinal mechanical moduli. However, these measurements do not address the primary feature of a TI model. Namely, it also is needed to describe the anisotropy in shear moduli.

Response 26: We agree that a complete description of the stiffness or compliance tensor of tissue requires shear moduli. This work is focused on anisotropy in longitudinal moduli. We showed that Brillouin measurements of the cornea are modeled well by the simple transverse isotropic model, which has clinical relevance because Brillouin microscopy can obtain this information *in vivo*. To our knowledge, there are no techniques suitable to measure the anisotropy in shear moduli *in vivo*.

Changes made: To clarify the limitations and potential clinical impact of this work better, we added a paragraph at the end of the Discussion section.

Corneal shape is primarily determined by its response to the mechanical load induced by intraocular pressure. The appropriate modulus describing deformation under tensile or inflation loading is the Young's modulus. In a nearly incompressible isotropic material, such as most soft tissue, the Young's modulus is simply related to the shear modulus.

Response 27: We agree that the deformation of the corneal shape is governed by shear modulus, not longitudinal modulus. Nonetheless, the longitudinal modulus can provide useful insights into the substructure of tissues, namely the stiffness of solid proteins and their concentration with respect to liquid water. Our previous work has revealed differences in longitudinal modulus between healthy and diseased corneas and detected changes following surgical treatments (e.g. [1], [2]).

[1] Shao, P. *et al.* Spatially-resolved Brillouin spectroscopy reveals biomechanical abnormalities in mild to advanced keratoconus in vivo. *Sci. Reports* 9, 7467 (2019).

[2] Eltony, A. M. *et al.* Brillouin Microscopy Visualizes Centralized Corneal Edema in Fuchs' Endothelial Dystrophy. *Cornea* 39 (2), 168-171 (2020).

Changes made: To clarify the limitations and potential clinical impact of this work better, we added a paragraph at the end of the Discussion section.

In a TI model, there are two shear moduli. The Young's modulus for a nearly incompressible TI material, such as the cornea, is primarily determined by one of the shear moduli and its response to torsion is primarily determined by the other shear modulus (see specific comment below). Consequently, measurements of anisotropic longitudinal elastic

moduli cannot be used to predict how the cornea will deform under mechanical loading but they can infer some underlying properties of the fiber matrix creating a TI structure.

Response 28: We fully appreciate the difference between longitudinal and shear moduli. Please see Responses 26 and 27 above.

Changes made: To clarify the limitations and potential clinical impact of this work better, we added a paragraph at the end of the Discussion section.

This paper highlights a fundamental issue with Brillouin microscopy for mechanical modeling of the cornea. There is no direct link between longitudinal and shear elastic moduli, especially in nearly incompressible materials such as soft tissue. Since the deformational properties of soft tissue are determined primarily by shear moduli, it is not clear at all if measurements of anisotropic longitudinal moduli will have any clinical relevance. The authors must address this point to properly judge the value of their contribution.

Response 29: Please see Responses 26 and 27 above.

Changes made: To clarify the limitations and potential clinical impact of this work better, we added a paragraph at the end of the Discussion section.

Specific Comments

Abstract, last sentence – The potential for clinical applications remains unclear since only longitudinal moduli are measured. This sentence should be qualified.

Response 30: In this work, we showed that Brillouin measurements are sensitive to the underlying fiber structure of the cornea. Hence, it is not unreasonable to predict potential detection of disruption or remodeling of the regular collagen lattice due to disease or surgical intervention, which certainly has clinical relevance in ophthalmology. Furthermore, our previous work has revealed differences in longitudinal modulus between healthy and diseased corneas and detected changes following surgical treatments (e.g. [1], [2]).

[1] Shao, P. *et al.* Spatially-resolved Brillouin spectroscopy reveals biomechanical abnormalities in mild to advanced keratoconus in vivo. *Sci. Reports* 9, 7467 (2019).

[2] Eltony, A. M. *et al.* Brillouin Microscopy Visualizes Centralized Corneal Edema in Fuchs' Endothelial Dystrophy. *Cornea* 39 (2), 168-171 (2020).

Introduction, end of last paragraph – There has been long-term interest in understanding the deformation of soft tissue under mechanical loading. However, for a nearly incompressible material like soft tissue, deformational properties are almost exclusively determined by shear moduli, which are not the focus of this study.

Response 31: Thanks. We have revised the Introduction section to emphasize that Brillouin microscopy characterizes longitudinal moduli. We think that the ability to characterize microstructure, particularly collagen fibrils, will be useful to gain insight into shear modulus.

Changes made: We have revised the Introduction section to clarify this point.

Introduction, 2nd paragraph, 3rd sentence – The Brillouin frequency shift is related to the longitudinal elastic properties of the material. This should be explicitly noted.

Response 32: Added to the revised manuscript.

Changes made: The sentence was revised to clarify this.

Results, Composite model of corneal anisotropy – This model is well argued and makes physical sense in describing the expected longitudinal mechanical properties of the cornea.

Response 33: Thanks.

Pg. 4, Eqs. 4 and 5 – It's unfortunate that the authors chose the symbol ϵ to characterize the anisotropy parameter. It may be confused with the strain since ϵ is often used as the symbol for strain in continuum mechanics. The authors should consider changing this notation to make it very clear that these equations are the anisotropy parameter.

Response 34: We originally chose the symbol ϵ by analogy to the Thomsen anisotropic parameters. However, we agree this could be confusing so have changed this notation in the revised manuscript.

Changes made: We changed the anisotropy parameter symbol from ϵ to α throughout the manuscript.

Figure 2 – These results clearly show corneal anisotropy and the lamellar structure. They strongly support a TI model.

Response 35: Thanks.

Figure 3, minor point – The angle in tissue is compensated for refraction at the air-cornea interface, but is the same tissue probed at different angles? Doesn't the beam probe different sections of the cornea for different angles due to refraction?

Response 36: Thanks for this question. Yes, because of refraction at the air-cornea interface, we expect the region of tissue being probed at different angles to be slightly offset. Assuming that the properties of the tissue are mostly uniform within a local area, we don't expect this to have a significant effect on the results.

Changes made: We added this detail in Methods, section "Brillouin angle-dependence measurements".

Figure 4, minor point – Again, the angle is changed with scanning, but isn't a different section of the cornea probed at each angle? Does this affect measurement variability?

Response 37: Yes, during scanning a different lateral position is probed at each angle. As suggested by Reviewer 1, we performed additional experiments to investigate the variation across the cornea. We measured the Brillouin angle-dependence at different locations in porcine corneas and described our findings in the revised manuscript. Although there was some variation from one cornea to another, we did not observe a significant difference in the average fitted anisotropic parameter in the central cornea vs. in other locations. We also created Brillouin maps of porcine corneas, similar to the human maps, and included this new data in the revised manuscript. Analyzing the angle-dependence from these porcine maps to estimate $\alpha^{(\text{stroma})}_{xx}$, we found good agreement with the anisotropy parameter ($\alpha^{(\text{stroma})}_{xx}$) extracted from porcine measurements at a single-point.

Changes made: Fig. 4 has been updated with new data, along with the corresponding Results section, "Angle-dependence of bulk cornea Brillouin measurements". We have also added a new figure (Fig. 6) and a new section under Results, "Anisotropy in Brillouin maps of the porcine cornea", which present 2D Brillouin maps of porcine corneas (similar to the human maps).

General comment on Figures 3 and 4 - How are Brillouin measurements in soft tissue validated? Is there a gold standard with known longitudinal anisotropy that can be used to compare to corneal measurements to validate the findings?

Response 38: The longitudinal moduli of many solids and liquids are known, and our system was calibrated during every single measurement using reference materials with temperature corrected curves. Furthermore, during this

revision, we integrated a high-speed electro-optic modulator to calibrate the frequency shift very accurately in the 5-6 GHz range. Our current Brillouin system may be considered a gold standard.

Table 1 – The findings in this table are generally consistent with ultrasound speed measurements probing longitudinal moduli in soft tissue.

Response 39: We agree that the measured values are generally consistent with the literature data obtained by ultrasound speed measurement. We'd like to point out that Brillouin measurement has higher spatial resolution and precision (<0.1%) than ultrasound measurement.

Discussion, 1st paragraph, last two sentences – These conclusions are speculative. Detailed numerical simulations of the cornea focus on tissue deformations under applied loads. Such deformations are primarily determined by the Young's and shear moduli. In the limit of an incompressible material, both the Young's modulus and shear modulus are derived from the material parameters labeled G_{yz} and G_{xy} . That is, the primary moduli describing deformation of the cornea under applied load (e.g, intraocular pressure) cannot be directly identified from the longitudinal moduli. In particular, the Young's modulus is approximately $3G_{xy}$ and is not captured at all by the Brillouin measurements reported here. The authors need to greatly justify the statement that Brillouin measurements can affect clinical understanding when they cannot be directly used to predict the most important feature of mechanical modeling.

Response 40: We agree with this remark. Please see our responses above on this matter. We believe that our revision addressed this issue clearly and describes the potential clinical relevance of longitudinal modulus measurement.

Discussion, 2nd paragraph, last sentence – Again, needs to be justified, similar to comments on the 1st paragraph in this section.

Response 41: Thanks. We have revised this sentence in the Discussion section to emphasize that the longitudinal moduli are being probed.

Discussion, 3rd paragraph, last sentence – Again, needs to be justified, similar to comments on the 1st paragraph in this section.

Response 42: In this case, we mentioned that Brillouin microscopy "could provide valuable material data", which is not the same thing as saying that it will provide comprehensive material data or the entirety of the material data that would be of interest. Especially in the revised version, we have emphasized the basic fact that Brillouin microscopy measures longitudinal moduli and not all moduli, so, we believe this sentence is sufficiently qualified.

Discussion, 4th paragraph, last sentence – Again, needs to be justified, similar to comments on the 1st paragraph in this section.

Response 43: Thanks. Revised.

Changes made: To clarify the limitations and potential clinical impact of this work better, we added a paragraph at the end of the Discussion section.

The discussion section focuses on the potential applications of Brillouin microscopy to predict the biomechanical behavior of many types of soft tissue. This has to be justified given that deformations are primarily determined by shear characteristics and Brillouin microscopy probes longitudinal characteristics.

Response 44: Please see our responses above on this issue.

Methods, Cross-sectional Brillouin measurements – These measurements have high spatial resolution, as noted, but take a very long time to perform. How practical will they be for clinical applications?

Response 45: The spectral acquisition time per location is typically a few seconds. In NIH-funded human studies, over 300 patients have been scanned using this instrument, generating useful data. Going forward towards clinical translation, efforts to increase the measurement speed are certainly warranted. This is an ongoing area of research in several labs.

Methods, Brillouin maps of human subjects –These measurements are not high spatial resolution and still take over 10 seconds. How practical can they be for general clinical use?

Response 46: Although each map is constructed from discrete locations, at each location the Brillouin data represents local tissue properties within a small volume. Again, several labs around the world are developing Brillouin microscopy for ophthalmic applications and using it to investigate the clinical potential of this technology, for example, for disease diagnosis and treatment monitoring.

Supplementary Materials – Eq. 11 – See note above about notation. The authors should consider changing this notation to make it very clear that these equations are the anisotropy parameter.

Response 47: Thanks. We changed the symbol from ϵ to α throughout the manuscript.

REVIEWERS' COMMENTS

Reviewer #1 (Remarks to the Author):

The authors addressed all my previous suggestions and concerns satisfactorily.

Reviewer #2 (Remarks to the Author):

The revised manuscript addresses all of the significant issues raised during the review of the initial submission. Additional measurements have been included, scientific notation has been changed to make it easier to follow all derivations, and assertions have been qualified to make the specific contributions of this paper very clear. Overall, the revised manuscript is significantly improved over the original.

There is one point the authors should address in finalizing the manuscript for publication. In the discussion section near the top of page 11, the authors state:

"Unfortunately, at present there are no instruments measuring anisotropy in shear moduli in vivo directly, though developing technologies, such as optical coherence elastography, may enable anisotropy measurement in shear moduli⁴⁶."

I believe that the authors should note more recent work on measuring elastic anisotropy in the cornea using optical coherence elastography (OCE) that clearly shows the large anisotropy in shear properties that dominates deformations. These OCE measurements have been confirmed in an animal model using destructive mechanical testing. Please note the two recent publications listed below.

JJ Pitre, MA Kirby, DS Li, TT Shen, RK Wang, M O'Donnell, I Pelivanov, "Nearly-incompressible transverse isotropy (NITI) of cornea elasticity: model and experiments with acoustic micro-tapping OCE," *Scientific Reports* 10(1), pp. 1-14, <https://doi.org/10.1038/s41598-020-69909-9> (2020).

MA Kirby, JJ Pitre, Jr., H-C Liou, DS Li, RK Wang, I Pelivanov, M O'Donnell, TT Shen, "Delineating Corneal Elastic Anisotropy in a Porcine Model Using Noncontact OCT Elastography and Ex Vivo Mechanical Tests," *Ophthalmology Science* 1(4), <https://doi.org/10.1016/j.xops.2021.100058> (2021).

POINT BY POINT RESPONSE TO REVIEWER COMMENTS

Reviewer #2 (Remarks to the Author):

The revised manuscript addresses all of the significant issues raised during the review of the initial submission. Additional measurements have been included, scientific notation has been changed to make it easier to follow all derivations, and assertions have been qualified to make the specific contributions of this paper very clear. Overall, the revised manuscript is significantly improved over the original.

There is one point the authors should address in finalizing the manuscript for publication. In the discussion section near the top of page 11, the authors state:

"Unfortunately, at present there are no instruments measuring anisotropy in shear moduli in vivo directly, though developing technologies, such as optical coherence elastography, may enable anisotropy measurement in shear moduli [46]."

I believe that the authors should note more recent work on measuring elastic anisotropy in the cornea using optical coherence elastography (OCE) that clearly shows the large anisotropy in shear properties that dominates deformations. These OCE measurements have been confirmed in an animal model using destructive mechanical testing. Please note the two recent publications listed below.

JJ Pitre, MA Kirby, DS Li, TT Shen, RK Wang, M O'Donnell, I Pelivanov, "Nearly-incompressible transverse isotropy (NITI) of cornea elasticity: model and experiments with acoustic micro-tapping OCE," *Scientific Reports* 10(1), pp. 1-14, <https://doi.org/10.1038/s41598-020-69909-9> (2020).

MA Kirby, JJ Pitre, Jr., H-C Liou, DS Li, RK Wang, I Pelivanov, M O'Donnell, TT Shen, "Delineating Corneal Elastic Anisotropy in a Porcine Model Using Noncontact OCT Elastography and Ex Vivo Mechanical Tests," *Ophthalmology Science* 1(4), <https://doi.org/10.1016/j.xops.2021.100058> (2021).

Response: Thank you for bringing this recent OCE work to our attention. It is certainly relevant here. We have added a comment about this to the *Discussion* section, including these two citations (highlighted in red in the revised manuscript).